# Continuous Treatment Effect Estimation with Cauchy-Schwarz Divergence Information Bottleneck

**Louk van Remmerden**                                              *l.smalbil@vu.nl*
*Department of Computer Science*
*Vrije Universiteit Amsterdam*

**Shiqin Tang**                                        *shiqin.tang@cair-cas.org.hk*
*Centre for Artificial Intelligence and Robotics*
*Hong Kong Institute of Science & Innovation, Chinese Academy of Sciences*

**Shujian Yu**                                                      *s.yu3@vu.nl*
*Department of Computer Science*
*Vrije Universiteit Amsterdam*

**Reviewed on OpenReview:** *https://openreview.net/pdf?id=9SvYOmMr2u*

## Abstract

Estimating conditional average treatment effects (CATE) for continuous and multivariate treatments remains a fundamental yet underexplored problem in causal inference, as most existing methods are confined to binary treatment settings. In this paper, we make two key theoretical contributions. First, we derive a novel counterfactual error bound based on the Cauchy–Schwarz (CS) divergence, which is provably tighter than prior bounds derived from the Kullback–Leibler (KL) divergence. Second, we strengthen this bound by integrating the Information Bottleneck principle, introducing a compression regularization on latent representations to enhance generalization. Building on these insights, we propose a new neural framework that operationalizes our theory. Extensive experiments on three benchmarks show that our method consistently outperforms state-of-the-art baselines and remains robust under biased treatment assignments.

## 1 Introduction

Estimating individual causal effects from observational data is inherently difficult because counterfactual outcomes are never observed, making direct validation impossible (Imbens & Rubin, 2015). As such, it has been recognized that robust causal inference methods must exhibit strong theoretical properties to ensure that treatment-effect estimates are well-bounded (Shalit et al., 2017).

In the binary treatment setting (where a certain treatment is either prescribed or not prescribed), the main objective of much of the theoretical work is to achieve group rebalancing. Group rebalancing is a procedure which essentially aimed to counter any bias that may occur in observational data due to, for instance, treatment effect bias (e.g., certain groups of patients may receive more treatment $T$ than others). As such, reducing the effect of confounding is a key step toward obtaining more reliable estimates from observational data. (Holland, 1986; Rubin, 2005).

While traditional causal inference methods such as inverse propensity score weighting (IPW) Austin & Stuart (2015) to directly re-weight the impact of treatment of the outcome, deep causal machine learning approaches, such as counterfactual regression methods (e.g., Shalit et al. (2017)), achieve bias reduction by learning a shared representation that balances the treatment groups in representation space. Under certain assumptions, it then becomes possible to derive generalization bounds on the conditional average treatment effect (CATE) estimates, limiting potential error and improving performance (Bellot et al., 2022; Shalit

et al., 2017). The key idea behind these methods is that, by using distributional measures such as integral probability metrics (IPM), one can quantify the distributional shift between the treated and control groups.

Extending counterfactual regression with distributional methods paradigm to the continuous treatment setting, where treatments represent real-valued dosages or multivariate exposures, introduces additional complexity. First, CATEs are now functions over a continuum of treatments, not binary, discrete units. Second, confounding adjustment requires estimating generalized propensity *densities* rather than scores (Imbens, 2000). Finally, many existing architectures, such as DRNet (Schwab et al., 2020), scale poorly, requiring a separate output head per treatment stratum.

Recent work addresses some of these challenges using adversarial approaches (Bica et al., 2020; Kazemi & Ester, 2024), extending generative adversarial networks (GANs) for binary treatment effect prediction to the continuous setting. However, these approaches remain adaptations of binary-treatment methods, and they often struggle with instability, sensitivity to hyperparameters, or poor scalability to high-dimensional treatment spaces.

In contrast, *information bottleneck* (IB) methods offer a theoretically grounded alternative. IB aims to learn representations $Z$ of covariates $X$ that are maximally predictive of outcomes $Y$ while discarding irrelevant information in $X$ (Tishby et al., 2000). When applied to causal inference, IB has been shown to reduce confounding in binary settings (Parbhoo et al., 2020; Lu et al., 2022). Yet, despite their promising properties, there is a lack of strong theoretical analysis on how IB methods improve counterfactual generalization. Moreover, IB has never been applied to continuous or multivariate treatment effect estimation.

To explore the potential of IB in more complex treatment settings, we propose **I**nformation **B**ottleneck for **E**stimating continuous e**X**posures (**IBEX**), a novel framework for CATE estimation with continuous and multivariate treatments. IBEX minimizes the statistical dependence between the learned representation and the treatment variable using a tractable approximation of mutual information based on the Cauchy-Schwarz (CS) divergence (Yu et al., 2024). To further encourage invariance and generalization, we apply a dimensionality bottleneck to the latent space.

> **Contributions.** ① We derive novel *counterfactual generalization bounds* for continuous treatments using the CS divergence, and show that these bounds are *tighter* than those based on the Kullback–Leibler divergence under mild assumptions. ② We design a modular architecture with separate covariate and treatment encoders, incorporating dimensionality regularization to control representation capacity. ③ We empirically validate IBEX across three benchmarks (MIMIC-IV, TCGA, News), showing *state-of-the-art* performance in terms of dose-response estimation and policy regret, and robustness under strong treatment-assignment bias.

## 2 Related Work

In this section, we provide a brief overview of relevant prior work on treatment effect estimation, with a particular focus on continuous treatments and representation learning approaches.

### 2.1 Information Bottleneck and its Application to Causal Inference

The Information Bottleneck (IB) principle, introduced by (Tishby et al., 2000), formulates representation learning as a trade-off between extracting information from the input variable $\mathbf{x}$ that is relevant for predicting the target variable $y$, and discarding nuisance factors in $\mathbf{x}$ that are irrelevant to $y$. Formally, the objective of IB is to learn a compressed representation $\mathbf{z}$ by minimizing the mutual information $I(\mathbf{x}; \mathbf{z})$, ensuring the minimality and compactness of $\mathbf{z}$, while simultaneously maximizing $I(\mathbf{z}; \mathbf{y})$, thereby preserving the predictive information to $y$. The optimization objective can thus be written as:

$$\min \quad I(\mathbf{x}; \mathbf{z}) \; - \; \beta I(\mathbf{z}; y), \tag{1}$$

where $\beta > 0$ controls the trade-off between compression and prediction. It has been theoretically shown that $Z$ naturally constitutes the minimal sufficient representation (Gilad-Bachrach et al., 2003).

To make this objective tractable for high-dimensional data and deep learning models, variational approximations such as the variational information bottleneck (VIB) (Alemi et al., 2016) and the nonlinear information bottleneck (NIB) (Kolchinsky et al., 2019) have been proposed.

Recent works have applied the IB principle in discrete treatment settings (Kim et al., 2019; Parbhoo et al., 2020; Lu et al., 2022). In principle, these approaches share a common high-level idea: leveraging the IB principle to compress high-dimensional covariates $\mathbf{x}$ into a low-dimensional representation $\mathbf{z}$ that retains information relevant for treatment effects $\{y, t\}$. They typically formulate an IB objective of the form:

$$\min I(\mathbf{x}; \mathbf{z}) - \beta I(\mathbf{z}; y, t), \tag{2}$$

with some methodological differences. For instance, the causal effect information bottleneck (CEIB) (Parbhoo et al., 2020) learns discrete latent representations separately from $\mathbf{x}_0$ (covariates of untreated patients) and $\mathbf{x}_1$ (covariates of treated patients). In contrast, causal information bottleneck (CIB) (Kim et al., 2019) uses two separate heads to estimate $I(\mathbf{z}; y_0)$ and $I(\mathbf{z}; y_1)$, where $y_0$ and $y_1$ denote the control and treatment outcomes, respectively.

However, these methods are limited to binary treatments and rely heavily on variational approximations, which are known to suffer from loose bounds and biased estimates of mutual information.

In this paper, we extend the IB framework to continuous treatment effect estimation by introducing a novel IB objective that differs fundamentally from Eq. (2). What is more, our implementation avoids variational approximation entirely, thereby mitigating the issues of bound looseness and biased information estimation. Moreover, we provide a formal analysis of the generalization error bound, which, to the best of our knowledge, has not been addressed in prior IB-based works.

## 2.2 Continuous Treatment Effect Estimation

Recent work has focused on extending methods originally developed for binary treatment estimation to the more realistic setting of continuous treatments, where dosages or combinations of dosages are considered. Bellot et al. (2022) derive generalization bounds for continuous treatment effect estimation and make use of a HSIC-type regularization, extending prior literature. Tanimoto et al. (2021) propose a regret-minimization approach for handling large action spaces. Schweisthal et al. (2023) develop a conformal prediction framework for estimating generalized propensity scores. Schwab et al. (2020) introduce DRNet, a representation learning method inspired by counterfactual regression approaches such as Shalit et al. (2017). Another notable line of work using conformal prediction for continuous treatments is presented by Schröder et al. (2024). Bica et al. (2020) (SCIGAN) and Kazemi & Ester (2024) (ACFR) adopt adversarial approaches to address the intractability of posteriors and use a Kullback–Leibler regularizer to correct for distributional shift. In contrast, we aim to improve robustness by leveraging the more stable Cauchy–Schwarz divergence.

## 3 Theoretical Preliminaries

The objective of our approach is to estimate individualized treatment effects (ITE) in settings with multivariate, continuous treatments, such as personalized dosage recommendations in healthcare. In this section, we introduce the relevant formalizations.

**Terminology.** Let $\mathcal{D}_{\mathrm{f}}$ be the factual dataset that contains i.i.d. samples $(\mathbf{x}^i, \mathbf{t}_{\mathrm{f}}^i, y_{\mathrm{f}}^i)$ drawn from distribution $p_{\mathbf{X}, \mathbf{T}_{\mathrm{f}}, y_{\mathrm{f}}}$. Let $\mathbf{X}$ denote a covariate vector taking values $\mathbf{x} \in \mathcal{X}$ (e.g. age, weight, lab results), and $\mathbf{x}$ represents a realization of $\mathbf{X}$. The treatment variable is in the form $\mathbf{T}_{\mathrm{f}} = (W_{\mathrm{f}}, D_{\mathrm{f}}) \in \mathcal{T}$, where the discrete component $W_{\mathrm{f}} \in \mathcal{W} = \{w_1, \ldots, w_k\}$ denotes the treatment type (e.g. specific combination of medications) and $D_{\mathrm{f}} \in \mathcal{D}_{W_{\mathrm{f}}}$ denotes the associated dosage (e.g. a number in $[0, 1]$ indicating the amount of medication provided). We denote the factual outcome as $Y_{\mathrm{f}} = Y(\mathbf{T}_{\mathrm{f}})$ and the counterfactual (i.e. unobserved) outcome as $Y_{\mathrm{cf}}$.

While there is only one pair of counterfactual treatment and outcome under the binary treatment setting, there are infinitely many of them in the case of continuous treatment. Therefore, we define the individual dose-response function.

**Definition 1.** *For any covariate vector $\mathbf{x} \in \mathcal{X}$, we define the dose–response function as*

$$\mu(\mathbf{t}, \mathbf{x}) := \mathbb{E}[Y(\mathbf{t})|\mathbf{X} = \mathbf{x}], \qquad \forall \mathbf{t} \in \mathcal{T}. \tag{3}$$

**Definition 2.** *The generalized propensity score (Imbens, 2000) is given by the conditional density*

$$e(\mathbf{x}) := p_{\mathbf{T}_f|\mathbf{X}}(\mathbf{t}|\mathbf{x}), \qquad \forall \mathbf{x} \in \mathcal{X}, \tag{4}$$

*where $\mathbf{t}$ may contain a continuous component.*

The generalized propensity score generalizes the conventional propensity score to account for continuous treatment.

**Definition 3.** *We define the treatment effect for a treatment-effect pair $t_1, t_2 \in \mathcal{T}$ as*

$$\tau_{t_1,t_2}(\mathbf{x}) := \mu(\mathbf{x}, t_1) - \mu(\mathbf{x}, t_2), \qquad \forall \mathbf{x} \in \mathcal{X}. \tag{5}$$

Equation equation 5 measures the relative treatment effect between different medications administered to the same subject.

**Assumption 1** (Ignorability Assumption). *We assume that the potential outcome is independent from the treatment given the sufficient adjustment set $\mathbf{X}$, i.e. $\mathbf{X}$ blocks all non-causal paths between treatment and outcome,*

$$\big\{Y(w, d)\big\}_{(w,d)\in\mathcal{T}} \perp\!\!\!\perp \mathbf{T}_f \mid \mathbf{X}. \tag{6}$$

**Assumption 2** (Overlap Assumption). *We assume that the conditional treatment distribution admits a density $p_{\mathbf{T}_f|\mathbf{X}}(\mathbf{t} \mid \mathbf{x})$ with respect to the Lebesgue measure, and that this density is strictly positive almost everywhere on $\mathcal{X} \times \mathcal{T}$, i.e.,*

$$p_{\mathbf{T}_f|\mathbf{X}}(\mathbf{t} \mid \mathbf{x}) > 0 \quad \text{for almost every } (\mathbf{x}, \mathbf{t}) \in \mathcal{X} \times \mathcal{T}. \tag{7}$$

Going forward, we introduce a stochastic encoder $q_\phi$ which compresses the covariate space $\mathcal{X}$ into a low-dimensional latent space $\mathcal{Z}$ and a predictor model $f : \mathcal{Z} \times \mathcal{T} \to \mathcal{Y}$. Additionally, let $L : \mathcal{Y} \times \mathcal{Y} \to \mathbb{R}^+$ be a loss function.

**Definition 4.** *Define the unit loss $\ell_{L,f,\phi} : \mathcal{X} \times \mathcal{T} \to \mathbb{R}^+$ as*

$$\ell_{L,f,\phi}(\mathbf{x}, \mathbf{t}) = L(f(\phi(\mathbf{x}), \mathbf{t}), y). \tag{8}$$

*Unit loss $\ell_{L,f,\phi}(\mathbf{x}, \mathbf{t})$ measures the loss between the predicted outcome $\hat{y} = f(\phi(\mathbf{x}), \mathbf{t})$ and the ground-truth outcome $y = \mu(\mathbf{x}, \mathbf{t})$.*

**Definition 5.** *We define the factual and counterfactual errors at treatment $t \in \mathcal{T}$ respectively as*

$$\epsilon_f^\ell(\mathbf{t}) := \int_{\mathcal{X}} \ell_{L,f,\phi}(\mathbf{x}, \mathbf{t}) \, p(\mathbf{x}|\mathbf{t}) \, d\mathbf{x}, \tag{9}$$

$$\epsilon_{cf}^\ell(\mathbf{t}) := \int_{\mathcal{T}'=[0,1]\setminus\{\mathbf{t}\}} \int_{\mathcal{X}} \ell_{L,f,\phi}(\mathbf{x}, \mathbf{t}) \, p(\mathbf{x}|\mathbf{t}') \, d\mathbf{x} \, d\mathbf{t}'$$
$$= \int_{\mathcal{X}} \ell_{L,f,\phi}(\mathbf{x}, \mathbf{t}) \, p(\mathbf{x}) \, d\mathbf{x}. \tag{10}$$

*Essentially, factual error $\epsilon_f^\ell(\mathbf{t})$ is obtained by marginalizing over $p(\mathbf{x}|\mathbf{t})$ while the counterfactual error $\epsilon_{cf}^\ell(\mathbf{t})$ is obtained by marginalizing over $p(\mathbf{x})$. Furthermore, we define $\epsilon_f = \int_{\mathcal{T}} \epsilon_f^\ell(\mathbf{t}) p(\mathbf{t}) d\mathbf{t}$ and $\epsilon_{cf} = \int_{\mathcal{T}} \epsilon_{cf}^\ell(\mathbf{t}) p(\mathbf{t}) d\mathbf{t}$.*

**Definition 6** (Cauchy-Schwarz Divergence (Principe et al., 2000; Jenssen et al., 2006)). *Let $\mu, \nu \in \mathcal{M}_+^1(\mathcal{X})$ be probability measures on a Borel subset $\mathcal{X} \in \mathbb{R}^d$. Assume $\mu$ and $\nu$ are absolutely continuous with respect to the Lebesgue measure Leb, and denote their density functions by $p = d\mu/d\text{Leb}$ and $q = d\nu/d\text{Leb}$. If $p, q \in L^2(\text{Leb})$, then Cauchy-Schwarz inequality gives*

$$\left(\int_{\mathcal{X}} p(\mathbf{x})q(\mathbf{x}) \, d\mathbf{x}\right)^2 \leq \left(\int_{\mathcal{X}} p^2(\mathbf{x}) \, d\mathbf{x}\right)\left(\int_{\mathcal{X}} q^2(\mathbf{x}) \, d\mathbf{x}\right), \tag{11}$$

*with equality holding if and only if p and q are colinear, almost everywhere on $\mathcal{X}$.*

*The CS divergence defines the distance between p and q by measuring the tightness (or gap) of the two sides of Eq. equation 11 using the logarithm of their ratio:*

$$D_{CS}(p\|q) = -\log\left(\frac{\left(\int p(\mathbf{x})q(\mathbf{x})\,d\mathbf{x}\right)^2}{\int p(\mathbf{x})^2\,d\mathbf{x} \cdot \int q(\mathbf{x})^2\,d\mathbf{x}}\right). \tag{12}$$

The CS divergence possesses several appealing properties compared to the KL divergence, which is defined as

$$D_{\mathrm{KL}}(p\|q) = \int p(\mathbf{x})\log\frac{p(\mathbf{x})}{q(\mathbf{x})}d\mathbf{x}. \tag{13}$$

In particular, the CS divergence is symmetric and admits closed-form expressions for mixture-of-Gaussians (MoG) distributions (Tran et al., 2022). In the following subsection, we further demonstrate how the use of CS divergence leads to a tighter generalization error bound compared to its KL divergence counterpart.

## 4 Continuous Treatment Generalization Bounds

We begin by analyzing generalization in the continuous, multivariate treatment setting. Under standard regularity conditions, we derive a counterfactual error bound in terms of the CS divergence between the learned representation and the treatment variable. Compared with recent approaches (Bellot et al., 2022; Kazemi & Ester, 2024), the resulting bound is expressed directly in representation space and admits a tighter dependence characterization under multivariate treatments.

Beyond dependence control, we further investigate the role of representation capacity in generalization. In particular, we show that IB–inspired regularization provides an upper bound on the factual generalization error through capacity control.

Taken together, this section provides theoretical insights into two complementary aspects of counterfactual learning: (i) representation–treatment dependence, which governs counterfactual robustness; and (ii) representation capacity, which governs generalization from empirical to population risk. These two perspectives are analyzed separately and later combined to motivate the IBEX objective.

### 4.1 Bounding the Counterfactual Error via CS Divergence–Induced Regularization

Before presenting our main theoretical result, we first introduce Assumptions 3 and 4, following the framework of Kazemi & Ester (2024).

**Assumption 3.** *The encoder function $\phi : \mathcal{X} \to \mathcal{Z}$ is a twice-differentiable bijection. The representation space $\mathcal{Z}$ is the image of $\mathcal{X}$ under $\phi$ with the induced distribution $p_\phi(\mathbf{z})$.*

Assumption 3 is introduced as a technical condition to facilitate the density transformation and change-of-variable arguments required for deriving the counterfactual generalization bound in this section. In particular, the twice-differentiable bijection ensures that the induced distribution $p_\phi(\mathbf{z})$ is well-defined and allows tractable analysis of divergence terms Kazemi & Ester (2024).

Importantly, this assumption is not imposed in the practical model. When introducing the IB-inspired capacity-control regularization, we explicitly relax the bijectivity requirement and allow non-invertible encoders. In this relaxed regime, the IB regularization term $I(\mathbf{x}; \mathbf{z})$ is no longer constant and becomes finite and controllable Kawaguchi et al. (2023), making it a meaningful measure of representational capacity.

**Assumption 4.** *Let $G$ be a class of functions with infinity norm less than 1, $G = \{g : \mathcal{Z} \times \mathcal{T} \to \mathbb{R}^+ \mid \|g\|_\infty \leq 1\}$. Then, there exists a constant $C > 0$ such that*

$$\frac{\ell_{L,g,\phi}(\mathbf{x}, \mathbf{t})}{C} \in G.$$

*This means for any* $(\mathbf{x}, \mathbf{t})$ *we have*

$$\frac{\ell_{L,g,\phi}(\mathbf{x}, \mathbf{t})}{C} \leq 1.$$

**Theorem 1** (Counterfactual Generalization Bound, Gaussian Scenario). *Let* $\phi$ *be an encoder* $\mathcal{X} \to \mathcal{Z}$, *and let* $f$ *be an outcome function* $\mathcal{Z} \times \mathcal{T} \to \mathcal{Y}$. *Assume that the joint distribution* $p(z, t)$ *follows a multivariate Gaussian distribution:*

$$p(\mathbf{z}, \mathbf{t}) \sim \mathcal{N}\left(\begin{bmatrix} \mu_z \\ \mu_t \end{bmatrix}, \Sigma_1\right), \quad where \quad \Sigma_1 = \begin{bmatrix} \Sigma_z & \Sigma_{z,t} \\ \Sigma_{z,t}^T & \Sigma_t \end{bmatrix}.$$

*Let* $\Sigma_2$ *denote the covariance matrix of the product of marginals* $p(\mathbf{z})p(\mathbf{t})$, *i.e., the case where* $\mathbf{z} \perp \mathbf{t}$. *Then,*

$$\Sigma_2 = \begin{bmatrix} \Sigma_z & 0 \\ 0 & \Sigma_t \end{bmatrix}.$$

*Under Assumptions 3 and 4, we have:*

$$\epsilon_{\mathrm{cf}} \leq \epsilon_{\mathrm{f}} + C\sqrt{2D_{CS}(p_\phi(\mathbf{z}, \mathbf{t}) \,\|\, p_\phi(\mathbf{z})p(\mathbf{t}))}, \tag{14}$$

*if*

$$\sum_{i=1}^{d} \log\left(\frac{2 + \lambda_i + 1/\lambda_i}{4}\right) \geq 4,$$

*where* $\lambda_i$ *is the i-th eigenvalue of*

$$\Sigma_2^{-1}\Sigma_1 = \begin{bmatrix} I & \Sigma_z^{-1}\Sigma_{z,t} \\ \Sigma_t^{-1}\Sigma_{z,t}^T & I \end{bmatrix}.$$

*Proof.* All the proofs can be found in Appendix B. □

The conditions in Theorem 1 are easy to satisfy. In our study, the joint dimension is $d = d_t + d_z = 130$, with treatment dimension $d_t = 2$ and latent dimension $d_z = 128$. Each term $\log\left(\frac{2+\lambda_i+1/\lambda_i}{4}\right)$ is non-negative, since $\lambda + 1/\lambda \geq 2$ for $\lambda \in (0, 1]$. Thus, when $d$ is large, the total sum easily exceeds the threshold. Even if most $\lambda_i$ values are close to 1 (indicating weak correlation between $z$ and $t$), a small number of moderately deviating values (e.g., $\lambda_i \leq 0.7$) are sufficient to push the sum above the bound.

In fact, Theorem 1 can be extended to general joint distribution $p(z, t)$ without assuming Gaussianity.

**Proposition 1.** *Let* $\phi$ *be an encoder mapping* $\mathcal{X} \to \mathcal{Z}$, *and let* $f$ *be an outcome function* $\mathcal{Z} \times \mathcal{T} \to \mathcal{Y}$. *Assume that* $p(\mathbf{z}, \mathbf{t})$ *is an arbitrary joint distribution. Then, we have*

$$\epsilon_{\mathrm{cf}} \lesssim \epsilon_{\mathrm{f}} + C\sqrt{2D_{CS}(p_\phi(\mathbf{z}, \mathbf{t}) \,\|\, p_\phi(\mathbf{z})p(\mathbf{t}))}, \tag{15}$$

*where* $\lesssim$ *denotes "less than or approximately equal to," and the precise conditions under which this inequality holds are discussed in Appendix B.*

Theorem 1 and Proposition 1 imply that reducing counterfactual error requires not only minimizing the factual error (which is intuitive) but also encouraging independence between $\mathbf{z}$ and $\mathbf{t}$, since $p(\mathbf{z}, \mathbf{t}) = p(\mathbf{z})p(\mathbf{t})$ if and only if $\mathbf{z} \perp \mathbf{t}$.

**Remark 1** (Tighter Bound). *A similar counterfactual error bound based on KL divergence is presented in (Bellot et al., 2022; Kazemi & Ester, 2024):*

$$\epsilon_{\mathrm{cf}} \leq \epsilon_{\mathrm{f}} + C\sqrt{2D_{KL}(p_\phi(\mathbf{z}, \mathbf{t}) \| p_\phi(\mathbf{z})p(\mathbf{t}))}. \tag{16}$$

*In general, there is no universal ordering between* $D_{CS}$ *and* $D_{KL}$ *for arbitrary distributions. However, under mild regularity conditions, KL divergence admits a lower bound in terms of CS divergence. Specifically, let*

$\mathbf{u} = [\mathbf{z}; \mathbf{t}] \in \mathbb{R}^{d_{\mathbf{z}}+d_{\mathbf{t}}}$ *and let* $p, q$ *be density functions defined on a bounded integration domain* $K \subset \mathbb{R}^{d_{\mathbf{z}}+d_{\mathbf{t}}}$ *with* $|K| < \infty$. *Assume* $p, q$ *are Riemann integrable on* $K$, *then (Yin et al., 2024):*

$$C_1\Big(D_{CS}(p; q) - \log |K| + 2 \log C_2\Big) \le D_{KL}(p; q), \tag{17}$$

*where* $C_1 := \int_K p(\mathbf{u}) \, d\mathbf{u} > 0$ *and* $C_2 := C_1 \left(\int_K p^2(\mathbf{u}) \, d\mathbf{u} \int_K q^2(\mathbf{u}) \, d\mathbf{u}\right)^{-1/4}$.

*Equivalently,*

$$D_{CS}(p; q) \le \frac{1}{C_1} D_{KL}(p; q) + \log |K| - 2 \log C_2. \tag{18}$$

*Substituting this relation into Eq. (15) yields:*

$$
\begin{aligned}
\epsilon_{\mathrm{cf}} &\lesssim \epsilon_{\mathrm{f}} + C \sqrt{2 D_{CS}(p_\phi(\mathbf{z}, \mathbf{t}) \,\|\, p_\phi(\mathbf{z}) p(\mathbf{t}))} \\
&\le \epsilon_{\mathrm{f}} + C \sqrt{2 \left(\frac{1}{C_1} D_{KL}(p_\phi(\mathbf{z}, \mathbf{t}) \| p_\phi(\mathbf{z}) p(\mathbf{t})) + \log |K| - 2 \log C_2\right)}.
\end{aligned}
\tag{19}
$$

*This establishes a theoretical comparison between the CS-based and KL-based formulations in the bounded continuous setting considered here. The constants* $C_1$ *and* $C_2$ *are distribution-dependent but finite under the stated assumptions.*

It is important to note that a tighter bound does not necessarily imply improved empirical performance; rather, it guarantees only that, under the assumptions of the theory, the Cauchy-Schwarz divergence yields a tighter theoretical upper bound on counterfactual error than the KL divergence.

We can further provide a bound in terms of the precision estimation of heterogeneous effects (PEHE), a metric commonly used in causal inference to measure the treatment-effect error (Hassanpour & Greiner, 2019; Shalit et al., 2017).

**Definition 7.** *We define the expected precision of estimating heterogeneous effect (PEHE) between treatment pairs* $\mathbf{t}_1, \mathbf{t}_2 \in \mathcal{T}$ *as*

$$
\begin{aligned}
\varepsilon_{pehe}(\mathbf{t}_1, \mathbf{t}_2) := \int_{\mathcal{X}} \Big[ &(\mu(\mathbf{x}, \mathbf{t}_1) - \mu(\mathbf{x}, \mathbf{t}_2)) \\
&- (f(\phi(\mathbf{x}), \mathbf{t}_1) - f(\phi(\mathbf{x}), \mathbf{t}_2)) \Big]^2 p(\mathbf{x}) \, d\mathbf{x}.
\end{aligned}
\tag{20}
$$

Following Proposition 1, we can then easily derive the following bound.

**Proposition 2** (PEHE Error Bound). *Given an encoder* $\phi$ *and outcome prediction function* $f$ *and a unit-loss function* $\ell_{L,f,\phi}(\mathbf{x}, \mathbf{t})$ *that satisfies Assumption 4 and its associated* $L$ *is squared error* $\|\cdot\|^2$, *the following inequality holds:*

$$
\begin{aligned}
\varepsilon_{pehe}(\mathbf{t}_1, \mathbf{t}_2) \le\ &\varepsilon_{\mathrm{f}}^\ell(\mathbf{t}_1) + \varepsilon_{\mathrm{f}}^\ell(\mathbf{t}_2) \\
&+ \sqrt{2 D_{CS}(p_\phi(\mathbf{z}) \| p_\phi(\mathbf{z}|\mathbf{t}_1))} + \sqrt{2 D_{CS}(p_\phi(\mathbf{z}) \| p_\phi(\mathbf{z}|\mathbf{t}_2))}.
\end{aligned}
\tag{21}
$$

The bound effectively states that for any pair of treatments $\mathbf{t}_1, \mathbf{t}_2$, minimizing the divergence between the marginal $p_\phi(\mathbf{z})$ and the conditionals $p_\phi(\mathbf{z}|\mathbf{t}_i)$ reduces the dependence of the learned representation $\mathbf{z} = \phi(\mathbf{x})$ on the treatment assignment. This encourages the encoder $\phi$ to learn a *balanced representation*—that is, one where the distribution of $\mathbf{z}$ is approximately invariant across treatment groups:

$$p_\phi(\mathbf{z} \mid \mathbf{t}_1) \approx p_\phi(\mathbf{z} \mid \mathbf{t}_2) \approx p_\phi(\mathbf{z}),$$

which implies $\mathbf{z} \perp \mathbf{t}$, i.e., independence of the representation from the treatment assignment.

## 4.2 Bounding the Factual Error with IB Approach

We now analyze how representation complexity affects generalization of factual risk.

**Theorem 2** (Information Bottleneck Bound on Factual Error (Kawaguchi et al., 2023))**.** *Let $\phi$ be a stochastic encoder mapping input $\mathbf{x}$ to a representation $\mathbf{z} = \phi(\mathbf{x})$, and let $\ell_{L,h,\phi}(\mathbf{x}, t)$ be a loss function that is L-Lipschitz and bounded in $[0, 1]$. Suppose the training data $\{(\mathbf{x}_i, t_i)\}_{i=1}^n$ are drawn i.i.d. from the joint distribution $p(\mathbf{x}, t)$, where $t \in [0, 1]$. Then, with probability at least $1 - \eta$, the expected factual error satisfies:*

$$\epsilon_{\mathrm{f}} \leq \hat{\epsilon}_{\mathrm{f}} + B\sqrt{\frac{I(\mathbf{x}; \mathbf{z})}{n}} + \frac{\delta}{\sqrt{n}},$$

*where $\hat{\epsilon}_{\mathrm{f}} = \frac{1}{n}\sum_{i=1}^n \ell_{L,h,\phi}(\mathbf{x}_i, t_i)$ is the empirical factual error, $B$ is a constant depending on the Lipschitz constant of the loss function, $\delta$ is a vanishing term (e.g., $\mathcal{O}(\sqrt{\log(1/\eta)}/n^{1/4})$) as $n \to \infty$.*

**Remark 2** (Effect of IB on Counterfactual Error)**.** *Beyond its classical role in controlling the generalization gap between empirical and factual risk, IB-inspired regularization may also indirectly reduce the counterfactual error $\epsilon_{\mathrm{cf}}$. In practice, by applying a capacity-control regularization motivated by the Information Bottleneck principle, the learned representation $\mathbf{z} = \phi(\mathbf{x})$ is encouraged to discard task-irrelevant or treatment-specific features that may not generalize across treatment regimes. According to the data processing inequality (Cover & Thomas, 2006), we have*

$$I(\mathbf{z}; t) \leq I(\mathbf{x}; \mathbf{z}). \tag{22}$$

*Since the counterfactual error bound includes a divergence term of the form (see Eq. (19))*

$$\epsilon_{\mathrm{cf}} \leq \epsilon_{\mathrm{f}} + C\sqrt{2D_{\mathrm{KL}}(p(\mathbf{z}, t)\|p(\mathbf{z})p(t))} = \epsilon_{\mathrm{f}} + C\sqrt{2I(\mathbf{z}; t)},$$

*reducing the dependence between the representation $\mathbf{z}$ and the treatment variable $t$ tightens the gap between factual and counterfactual error. While Assumption 3 assumes a bijective encoder for theoretical tractability, in practical implementation this condition is relaxed. In the non-invertible regime, IB-inspired regularization effectively reduces $I(\mathbf{x}; \mathbf{z})$, which in turn constrains $I(\mathbf{z}; t)$ via data processing inequality, thereby promoting treatment-invariant representations and improving counterfactual robustness.*

## 4.3 Complementary Roles of Dependence and Capacity

Theorem 1 and Theorem 2 address two distinct sources of error in counterfactual learning. First, Theorem 1 shows that counterfactual robustness is governed by the statistical dependence between the learned representation $\mathbf{z}$ and the treatment variable $\mathbf{t}$. In particular, reducing the divergence term $D_{\mathrm{CS}}(p(\mathbf{z}, \mathbf{t})\|p(\mathbf{z})p(\mathbf{t}))$ tightens the gap between factual and counterfactual error, thereby mitigating counterfactual shift.

Second, Theorem 2 establishes that the generalization of the factual risk depends on representation capacity. Controlling the mutual information $I(\mathbf{x}; \mathbf{z})$ limits representation complexity and improves the transition from empirical to population risk.

Although these two bounds are derived under different technical regimes: Theorem 1 assuming an invertible encoder for analytical tractability, and Theorem 2 allowing non-invertible capacity-limited encoders, they provide complementary design principles for practical model construction:

- Minimize representation-treatment dependence to reduce counterfactual shift;
- Control representation capacity to improve generalization.

In practice, our IBEX jointly optimizes a dependence penalty $D_{\mathrm{CS}}(p(\mathbf{z}, \mathbf{t})\|p(\mathbf{z})p(\mathbf{t}))$ motivated by the counterfactual bound, together with a capacity-control regularizer approximating $I(\mathbf{x}; \mathbf{z})$ motivated by the factual generalization bound. While we do not claim a single unified bound under non-invertible encoders, these independently established theoretical results jointly justify the structure of the IBEX objective.

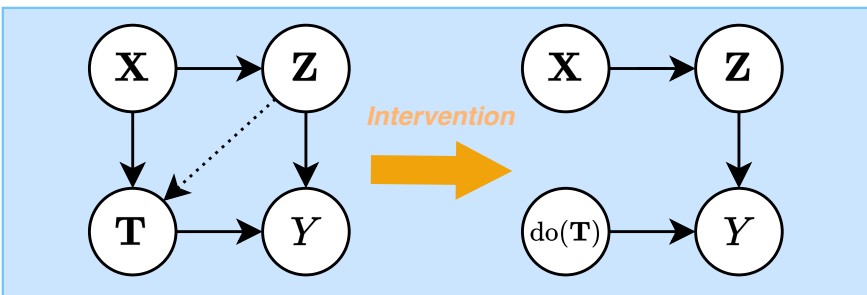

Figure 1: Structural causal models implementing the IBEX framework: pre-intervention scenario (left panel) and post-intervention outcome (right panel). The dotted arrow from **Z** to **T** indicates statistical dependence induced via the shared parent **X**. Note that **Z** is a function of **X** via $\mathbf{Z} = f(\mathbf{X})$.

## 5 Methodology

### 5.1 Structural overview

Before delving into the optimization objective, we first ground the reader in the structural causal model (SCM) that motivates IBEX.

Figure 1 contains the SCMs of the IBEX before and after intervention. In the conventional causal effect model, covariates **x** simultaneously influence the treatment $t$ and the outcome $y$, while $t$ also affects $y$. This layout conflates *all* information in **x** (both outcome–relevant and nuisance, treatment–specific factors), making generalisation difficult when the distribution of $t$ shifts.

On the other hand, IBEX inserts a learned bottleneck variable $\mathbf{z} = f_\phi(\mathbf{x})$ between **x** and the rest of the system and explicitly regularizes two information pathways: ① We maximize $I((\mathbf{z}, t); y)$ so that **z** keeps precisely the features of **x** needed, jointly with **t**, to predict $y$. ② We minimize $I(\mathbf{x}; \mathbf{z})$ and drive $I(\mathbf{z}; \mathbf{t})$ toward zero via a CS divergence term, forcing the encoder to forget treatment-specific components that do not help predict $y$.

### 5.2 Objective function and Implementation

Guided by the complementary principles established in Section 4.3, IBEX is formulated as an information-theoretic objective that simultaneously promotes predictive sufficiency, limits representation capacity, and reduces representation–treatment dependence. The high-level objective is:

$$\max \quad I\big((\mathbf{z}, t); y\big) \; - \; \beta I(\mathbf{x}; \mathbf{z}) \; - \; \gamma I(\mathbf{z}; t), \tag{23}$$

where $\beta, \gamma > 0$ are trade-off hyperparameters.

The first term $I((\mathbf{z}, t); y)$ enforces *sufficiency*, ensuring that the representation retains information necessary for predicting the outcome jointly with the treatment. The second term $I(\mathbf{x}; \mathbf{z})$ controls representation capacity in the spirit of the IB principle. Reducing this mutual information encourages compression of **x** into a minimal predictive representation, thereby improving generalization as suggested by Theorem 2. The third term $I(\mathbf{z}; t)$ penalizes statistical dependence between the representation and treatment assignment. Minimizing this term mitigates treatment-specific confounding and reduces counterfactual shift, consistent with the dependence-based bound in Theorem 1.

In practice, the mutual information terms are approximated using tractable surrogates. Representation capacity is controlled either explicitly through dimensionality constraints on **z** (Tao et al., 2020) or via regularized mutual-information estimators. The dependence term $I(\mathbf{z}; t)$ is approximated using the CS divergence, which provides a stable and computationally efficient measure of representation–treatment dependence in multivariate settings.

### 5.2.1 Maximizing Expressiveness Term

We maximize $I(\mathbf{z}, \mathbf{t}); y)$, which states that the encoding-treatment pair needs to be expressive enough to predict the outcome $Y$. This is implemented via standard empirical risk minimization as (Kolchinsky et al., 2019):

$$\mathbb{E}_{(\mathbf{x}, \mathbf{t}, y) \sim p(\mathbf{x}, \mathbf{t}, y)} \left[ \left( y - f(\phi(\mathbf{x}), \tilde{\mathbf{t}}) \right)^2 \right], \tag{24}$$

where $\tilde{\mathbf{t}}$ is a learned treatment embedding from an embedding function $\tau : (w, d) \in \mathcal{T} \to \tilde{\mathcal{T}}$ and $f$ an output predictor head with the mapping $\tilde{\mathcal{T}} \times \mathcal{Z} \to \mathcal{Y}$. We estimate this expectation using the empirical mean squared error (MSE) over the training data via $\frac{1}{N} \sum_{i=1}^{N} \left( y_i - f(\phi(\mathbf{x}_i), \tilde{\mathbf{t}}_i) \right)^2$.

### 5.2.2 Treatment-Compression Term

Minimizing $I(\mathbf{z}; \mathbf{t})$ is achieved by estimating the CS divergence between $\mathbf{T}$ and $\mathbf{Z}$. Our approach is conceptually similar to existing HSIC-based methods (e.g., Bellot et al. (2022)) which enforce $\mathbf{t} \perp \mathbf{z}$, but replaces HSIC with the CS divergence. Formally, we minimize:

$$D_{\text{CS}}(p(\mathbf{z}, \mathbf{t}) \| p(\mathbf{z})p(\mathbf{t})) = - \log \left( \frac{\left( \int p(\mathbf{z}, \mathbf{t}) \, p(\mathbf{z})p(\mathbf{t}) \, d\mathbf{z}d\mathbf{t} \right)^2}{\int p(\mathbf{z}, \mathbf{t})^2 \, d\mathbf{z}d\mathbf{t} \cdot \int p(\mathbf{z})^2 p(\mathbf{t})^2 \, d\mathbf{z}d\mathbf{t}} \right). \tag{25}$$

Given a batch of data points $\{(\mathbf{x}_i, \mathbf{t}_i)\}_{i=1}^{N} \sim p(\mathbf{x}, \mathbf{t})$, we compute representations via a deterministic encoder $\mathbf{z}_i = \phi(\mathbf{x}_i)$. This induces joint samples $\{(\mathbf{z}_i, \mathbf{t}_i)\}_{i=1}^{N} \sim p(\mathbf{z}, \mathbf{t})$, which can be used to assess statistical dependence between $\mathbf{z}$ and $\mathbf{t}$. The empirical CS divergence can then be estimated as (Yu et al., 2024):

$$\begin{aligned}
\widehat{I}_{\text{CS}}(\mathbf{z}; \mathbf{t}) &= \log \left( \frac{1}{N^2} \sum_{i,j}^{N} K_{ij} Q_{ij} \right) + \log \left( \frac{1}{N^4} \sum_{i,j,q,r}^{N} K_{ij} Q_{qr} \right) - 2 \log \left( \frac{1}{N^3} \sum_{i,j,q}^{N} K_{ij} Q_{iq} \right) \\
&= \log \left( \frac{1}{N^2} \text{tr}(KQ) \right) + \log \left( \frac{1}{N^4} \mathbb{1}^T K \mathbb{1} \mathbb{1}^T Q \mathbb{1} \right) - 2 \log \left( \frac{1}{N^3} \mathbb{1}^T K Q \mathbb{1} \right),
\end{aligned} \tag{26}$$

where $\mathbb{1}$ is a $N \times 1$ vector of ones, and $K$ and $Q$ denote the Gram matrices for variables $z$ and $t$, respectively. Specifically, $K_{i,j} = \kappa(z_i, z_j)$ with $\kappa$ is a positive-definite kernel, such as the Gaussian RBF kernel. The second equality of Eq. (26) reduces the complexity to $\mathcal{O}(N^2)$.

Note that our empirical estimator of $I_{\text{CS}}$ is fully non-parametric and does not rely on any parametric distributional assumptions on $p(z, t)$, such as Gaussianity, even though our first theoretical result in Theorem 1 assumes such a form.

### 5.2.3 Approximating the Compression Term

To minimize $I(\mathbf{x}; \mathbf{z})$, we limit the latent space via a fixed low-dimensional $\mathbf{z} \in \mathbb{R}^d$ and apply regularization. Group sparsity and entropy-based penalties (e.g., log-det covariance) further reduce $\mathbf{z}$'s capacity (Dai et al., 2018; Kawaguchi et al., 2023; Tishby et al., 2000). In particular, we have regularization term

$$R_{\dim}(\mathbf{z}) := \|\mathbf{z}^\top\|_{2,1} + \kappa \log \det (\Sigma_z + \epsilon I), \tag{27}$$

where $\|\mathbf{z}^\top\|_{2,1} = \sum_{j=1}^{d} \left( \sum_{i=1}^{N} z_{ij}^2 \right)^{1/2}$ is the $(2, 1)$-norm promoting column sparsity, and $\Sigma_z = \frac{1}{N} \sum_{i=1}^{N} (z_i - \bar{z})(z_i - \bar{z})^\top$ is the empirical covariance matrix. $I$ denotes the identity matrix of dimension $d_z \times d_z$ with a scalar $\epsilon$ to ensure numerical stability, $\bar{z}$ the empirical mean of the batch and $\kappa$ a hyperparameter which takes on values in $[0, 1]$ and is used to balances the two terms.

### 5.3 Model Objectives and Architecture

Our model architecture builds upon existing methodologies such as VCNet Bellot et al. (2022), but separates the treatment embedding and covariate embedding layers to align with the modelling objectives and the

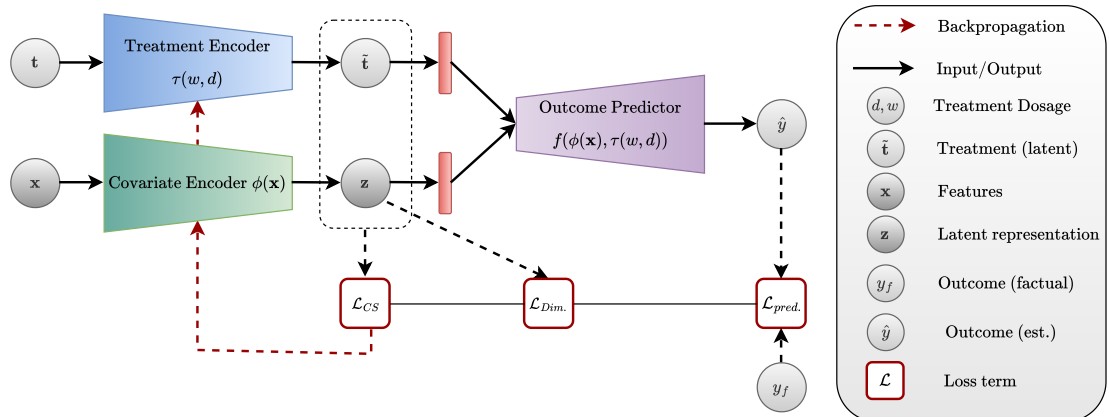

Figure 2: Illustration of the **IBEX** model architecture. IBEX jointly encodes treatment and covariate information using a *Treatment Encoder* $\tau(w, d)$ and a *Covariate Encoder* $\phi(x)$, where $w$ and $d$ denote the treatment identifier and dosage, respectively. The encoded representations are combined to predict the outcome $\hat{y}$ using an *Outcome Predictor* $f(\phi(x), \tau(w, d))$.

observational structure. As shown in Figure 2, we learn an embedding of the covariate space via $\phi : \mathcal{X} \to \mathcal{Z}$. A treatment encoder $\tau(w, d)$ learns a mapping $\mathcal{T} \to \tilde{\mathcal{T}}$ is used to impose a bottleneck in the architecture, promoting sample efficiency. Lastly, $f$ predicts outcomes given both $\tilde{\mathbf{t}}$ and the covariate representation $\mathbf{z}$. The network is optimized end-to-end using backpropagation, with gradients flowing through all components of the architecture.[1] The optimization objective consists of the empirical loss term, along with two theoretically motivated regularization terms. Formally:

$$
\mathcal{L}_{\text{IBEX}}(\phi, \tau, f) = \underbrace{\mathbb{E}_{(\mathbf{x}, \mathbf{t}, y) \sim p(\mathbf{x}, \mathbf{t}, y)} \left[ \left( y - f(\phi(\mathbf{x}), \tilde{\mathbf{t}}) \right)^2 \right]}_{\text{Prediction Term}}
$$
$$
+ \underbrace{\beta \cdot R_{\text{dim}}(\mathbf{z})}_{\text{Dimensionality Bottleneck}} + \underbrace{\gamma \cdot D_{\text{CS}} \left( p_\phi(\mathbf{z}, \mathbf{t}) \| p_\phi(\mathbf{z}) p(\mathbf{t}) \right)}_{\text{Treatment compression Term}},
$$
(28)

where $\beta$ and $\gamma$ are hyperparameters which take on values in $[0, 1]$.

## 6 Experiments

We conduct a series of experiments following the setups used in previous comparative studies Bellot et al. (2022); Bica et al. (2020); Kazemi & Ester (2024). We run the experiments on a MacOS M4 system with a 10-core CPU, 32 GB unified RAM, and 120 GB/s memory bandwidth. Our implementation is available at `https://github.com/ljsmalbil/IBEX`.

**Baselines.** We compare with: (1) **DRNet** (Schwab et al., 2020), with two variants—**HSIC** (Gretton et al., 2007) and **Wass** (Villani et al., 2008) regularization; (2) **SCIGAN** (Bica et al., 2020), a GAN-based counterfactual model; (3) Generalised Propensity Score **GPS** (Imbens, 2000); (4) a two-layer multilayer perceptron **MLP**; (5) **VCNet** (Bellot et al., 2022) with HSIC and Wass variants; (6) **ACFR** (Kazemi & Ester, 2024), which uses adversarial KL loss and attention; and (7) **GIKS** (Nagalapatti et al., 2024), which uses data-augmentation as a debiasing method.

**Benchmark Datasets.** We evaluate on three datasets: (1) **MIMIC-IV** (Johnson et al., 2023) contains records from 5,476 ICU patients who received mechanical ventilation. Treatment defined as a 2D continuous vector of ventilator settings (tidal volume and respiratory rate). (2) **News** (Asuncion et al., 2007) is a

---

[1]Note that, following standard practice in continuous causal modeling (e.g., Bellot et al. (2022), Kazemi & Ester (2024)), we assume $\phi$ to be a bijection for theoretical analysis. In practical implementation, however, $\phi$ is parameterized as a neural network and is not required to be invertible.

| Method | News | MIMIC-IV | TCGA |
|---|---|---|---|
| SCIGAN | $3.71 \pm 0.05$ | $2.09 \pm 0.12$ | $0.36 \pm 0.04$ |
| DRNet$_{\textbf{HSIC}}$ | $4.98 \pm 0.12$ | $4.45 \pm 0.07$ | $3.02 \pm 0.28$ |
| DRNet$_{\textbf{Wass}}$ | $5.07 \pm 0.12$ | $4.47 \pm 0.12$ | $1.73 \pm 0.26$ |
| VCNet$_{\textbf{HSIC}}$ | $3.41 \pm 0.11$ | $1.15 \pm 0.02$ | $0.95 \pm 0.02$ |
| VCNet$_{\textbf{Wass}}$ | $3.46 \pm 0.04$ | $1.23 \pm 0.12$ | $1.06 \pm 0.01$ |
| GPS | $6.97 \pm 0.11$ | $7.39 \pm 0.00$ | $6.12 \pm 0.91$ |
| MLP | $5.48 \pm 0.16$ | $5.34 \pm 0.16$ | $2.02 \pm 0.33$ |
| ACFR | $5.44 \pm 0.14$ | $2.19 \pm 0.19$ | $1.03 \pm 0.12$ |
| GIKS | $3.66 \pm 0.04$ | $\mathbf{1.09} \pm 0.06$ | $1.25 \pm 0.06$ |
| IBEX | $\mathbf{2.96} \pm 0.16$ | $\mathbf{1.05} \pm 0.02$ | $\mathbf{0.15} \pm 0.09$ |

Table 1: Out-of-sample performance of $\sqrt{\mathrm{MISE}}$ on News, MIMIC, and TCGA datasets. Lower is better. Highest performer ($p < 0.05$ paired t-test) in bold face. Results are averaged over $n = 10$ random seeds. We assess significance using paired two-sided t-tests between our method and each baseline, with Holm correction for multiple comparisons.

| Method | News | MIMIC-IV | TCGA |
|---|---|---|---|
| SCIGAN | $3.90 \pm 0.05$ | $0.32 \pm 0.05$ | $0.23 \pm 0.03$ |
| DRNet$_{\textbf{HSIC}}$ | $4.17 \pm 0.11$ | $1.44 \pm 0.05$ | $1.24 \pm 0.03$ |
| DRNet$_{\textbf{Wass}}$ | $4.56 \pm 0.12$ | $1.37 \pm 0.05$ | $1.27 \pm 0.05$ |
| VCNet$_{\textbf{HSIC}}$ | $3.10 \pm 0.21$ | $0.63 \pm 0.02$ | $0.39 \pm 0.01$ |
| VCNet$_{\textbf{Wass}}$ | $2.99 \pm 0.12$ | $0.58 \pm 0.03$ | $0.44 \pm 0.03$ |
| GPS | $24.1 \pm 0.55$ | $20.2 \pm 0.01$ | $1.26 \pm 0.01$ |
| MLP | $6.45 \pm 0.21$ | $1.65 \pm 0.05$ | $1.13 \pm 0.17$ |
| ACFR | $5.11 \pm 0.12$ | $0.80 \pm 0.02$ | $1.11 \pm 0.13$ |
| GIKS | $2.15 \pm 0.09$ | $0.51 \pm 0.02$ | $0.95 \pm 0.03$ |
| IBEX | $\mathbf{1.75} \pm 0.09$ | $\mathbf{0.31} \pm 0.04$ | $\mathbf{0.15} \pm 0.03$ |

Table 2: Out-of-sample performance of $\sqrt{\mathrm{PE}}$. Lower is better. Highest performer ($p < 0.05$ paired t-test) in bold face.

bag-of-words data set of New York Times articles. Lastly, (3) The Cancer Genome Atlas (**TCGA**) is a gene expression data for 9,000 cancer patients. We synthetically generate treatments and outcomes in accordance with previous work (e.g. Bica et al. 2020; Kazemi & Ester 2024; Schwab et al. 2020) (details in Appendix E).

**Evaluation metrics.** We report the square root of the **Mean Integrated Squared Error (MISE)**: $\frac{1}{N|\mathcal{W}|} \sum_{w \in \mathcal{W}} \sum_{i=1}^{N} \int_{\mathcal{D}_w} (y_i(w, d) - \hat{y}_i(w, d))^2 \, \mathrm{d}d$, which averages squared errors between true and predicted outcomes over individuals, treatment types $\mathcal{W}$, and dosage ranges $\mathcal{D}_w$ and effectively compares the true outcome for a given treatment (and dosage) and the predicted outcome. We also report the square root of the **Policy Error (PE)**: $\frac{1}{N} \sum_{i=1}^{N} (y_i(w_i^*, d_i^*) - y_i(\hat{w}_i^*, \hat{d}_i^*))^2$, where $(w_i^*, d_i^*)$ is the actual optimal treatment–dosage and $(\hat{w}_i^*, \hat{d}_i^*)$ is the predicted optimal treatment chosen by the model. PE quantifies regret from suboptimal policy choices.

## 6.1 Experimental Results

Tables 1-2 show that IBEX achieves the best performance on MIMIC with $\sqrt{\mathrm{MISE}} = 1.61$ and $\sqrt{\mathrm{PE}} = 0.12$, showing accurate outcome estimation and strong policy performance in a clinical setting with multivariate treatments, outperforming or matching VCNet-based approaches. On the News dataset, IBEX leads with the lowest $\sqrt{\mathrm{MISE}} = 2.96$ and $\sqrt{\mathrm{PE}} = 1.75$. On the TCGA dataset, IBEX again achieves the lowest ($\sqrt{\mathrm{MISE}} = 0.15$), indicating that our approach allows for precise modelling of gene expression outcomes and reliable treatment policy learning.

| Method | $\sqrt{\textbf{MISE}}$ | $\sqrt{\textbf{PE}}$ |
|---|---|---|
| IBEXDim. Reg. Only | $1.11 \pm 0.01$ | $0.57 \pm 0.01$ |
| IBEXCS. Reg. Only | $1.08 \pm 0.02$ | $0.59 \pm 0.02$ |
| IBEXNo regularization | $1.32 \pm 0.02$ | $0.71 \pm 0.01$ |
| IBEX | $1.05 \pm 0.16$ | $0.12 \pm 0.09$ |

Table 3: Out-of-sample performance on various versions of the IBEX model on the MIMIC-IV dataset. Lower is better. In this setting $\gamma = 0.01$ and $\beta = 0.1$.

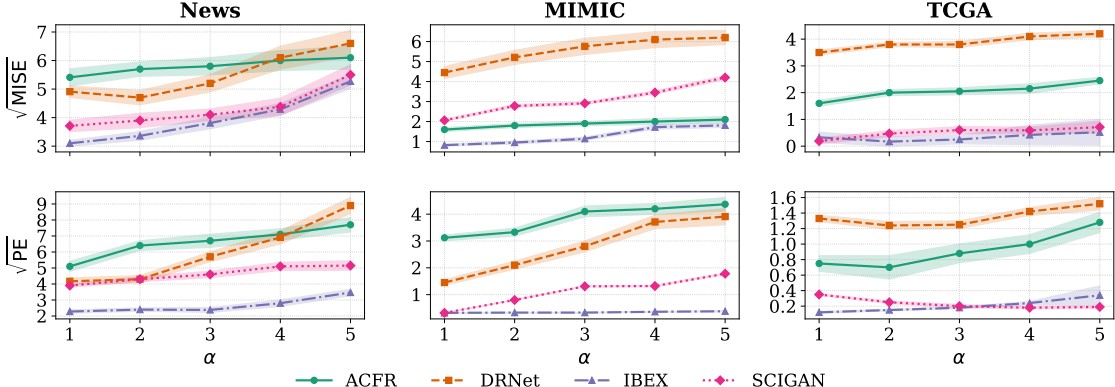

Figure 3: Results (out-of-sample) on News for Various Treatment-bias Values. We used DRNetHSIC in this setup.

**Ablation Results.** Table 3 reports MIMIC-IV results for four IBEX variants: using only the dimensionality regularizer ($\beta = 0.1$), only the Cauchy regularizer ($\gamma = 0.01$), neither regularizer, or both (full IBEX). Both regularizers independently improve performance over the unregularized baseline ($\sqrt{\text{MISE}} = 1.32$), with scores of 1.11 and 1.08, respectively, demonstrating the clear effectiveness of including the regularizers in the optimisation objective, as well as their complementary benefits.

**Treatment bias robustness.** We assess model robustness to treatment bias on the News and MIMIC-IV datasets. We vary the assignment bias $\alpha$ from 0 (none) to 10 (strong). As shown in Figure 3, IBEX maintains generally low $\sqrt{\text{MISE}}$ and $\sqrt{\text{PE}}$ even under strong bias, highlighting its resilience to confounding. In the MIMIC-IV setting on $\sqrt{\text{MISE}}$, ACFR also demonstrates a steady performance which eventually matches ours.

## 7 Discussion

IBEX outperforms all baselines across diverse datasets, demonstrating the strength of Cauchy–Schwarz (CS) divergence-based regularization for estimating continuous treatment effects. Its advantage is particularly evident under treatment-assignment bias, a common challenge in observational data. Beyond its theoretical grounding, IBEX performs well empirically across domains with different data characteristics, suggesting that the CS divergence provides a stable and expressive measure for enforcing representation invariance.

Nevertheless, like other causal inference approaches, IBEX relies on standard assumptions such as ignorability and overlap, which may not always hold in practice.

Future work could explore relaxing theoretical conditions, extending IBEX to dynamic treatment regimes (e.g., chronic care), and improving scalability. Integrating IBEX with conformal prediction or adversarial debiasing techniques may further enhance its utility in real-world settings with missing covariates or unmeasured confounding, broadening its impact in precision medicine and policy evaluation.

## 8 Conclusion

We introduced IBEX, a method for continuous and multivariate treatment effect estimation grounded in the information bottleneck principle. By deriving novel counterfactual generalization bounds and implementing them via two regularization objectives, IBEX improves out-of-sample performance while maintaining tractability, even with high-dimensional structured treatments.

### Acknowledgments

We would like to thank the reviewers and the action editor for their time and effort in improving this manuscript. This research was supported by the EU Horizon 2020 project (grant No. 96534, ICARE4OLD).

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

# A    Notations

| Symbol | Description |
|---|---|
| $\mathcal{D}_{\mathrm{f}}$ | Factual dataset with i.i.d. samples $(\mathbf{x}^i, \mathbf{t}_{\mathrm{f}}^i, y_{\mathrm{f}}^i)$ |
| $p_{\mathbf{x}, \mathbf{t}_{\mathrm{f}}, y_{\mathrm{f}}}$ | Joint distribution of covariates, treatments, and factual outcomes |
| $\mathbf{x} \in \mathcal{X}$ | Covariate vector (e.g., age, weight, lab results) |
| $\mathbf{t}_{\mathrm{f}} = (w_{\mathrm{f}}, d_{\mathrm{f}})$ | Factual treatment: type $w_{\mathrm{f}}$ and dosage $d_{\mathrm{f}}$ |
| $\mathcal{W}$ | Set of discrete treatment types $\{w_1, \ldots, w_k\}$ |
| $\mathcal{D}_w$ | Set of dosages for treatment type $w$ |
| $y_{\mathrm{f}} = Y(\mathbf{t}_{\mathrm{f}})$ | Factual outcome |
| $y_{\mathrm{cf}}$ | Counterfactual outcome |
| $\mu(\mathbf{t}, \mathbf{x})$ | Dose–response function: expected outcome under treatment $\mathbf{t}$ for covariates $\mathbf{x}$ |
| $e(\mathbf{x})$ | Generalized propensity score: $p(\mathbf{t}_{\mathrm{f}}\|\mathbf{x})$ |
| $\tau_{\mathbf{t}_1, \mathbf{t}_2}(\mathbf{x})$ | Individualized treatment effect between $\mathbf{t}_1$ and $\mathbf{t}_2$ |
| $\phi : \mathcal{X} \to \mathcal{Z}$ | Encoder mapping $\mathbf{x}$ to latent space $\mathbf{z}$ |
| $\tau : \mathcal{T} \to \tilde{\mathcal{T}}$ | Treatment mapping $\mathbf{t}$ to latent treatment space $\tilde{\mathbf{t}}$ |
| $f : \mathcal{Z} \times \tilde{\mathcal{T}} \to \mathcal{Y}$ | Outcome prediction function |
| $L : \mathcal{Y} \times \mathcal{Y} \to \mathbb{R}^+$ | Loss function (e.g., squared error) |
| $\ell_{L, f, \phi}(\mathbf{x}, \mathbf{t})$ | Unit loss between predicted and ground-truth outcome |
| $\epsilon_{\mathrm{f}}^{\ell}(\mathbf{t})$ | Factual error for treatment $\mathbf{t}$ |
| $\epsilon_{\mathrm{cf}}^{\ell}(\mathbf{t})$ | Counterfactual error for treatment $\mathbf{t}$ |
| $\epsilon_{\mathrm{f}}$ | Expected factual error over $p(\mathbf{t})$ |
| $\epsilon_{\mathrm{cf}}$ | Expected counterfactual error over $p(\mathbf{t})$ |
| $D_{\mathrm{CS}}(p\|q)$ | Cauchy–Schwarz divergence between distributions $p$ and $q$ |
| $D_{\mathrm{KL}}(p\|q)$ | Kullback–Leibler divergence between distributions $p$ and $q$ |
| $\mathbf{z} = \phi(\mathbf{x})$ | Latent representation of covariates |
| $\mathcal{Z}$ | Latent space (encoded covariate space) |
| $\Sigma_1$ | Covariance matrix of joint distribution $p(\mathbf{z}, \mathbf{t})$ |
| $\Sigma_2$ | Covariance matrix of product of marginals $p(\mathbf{z})p(\mathbf{t})$ |
| $\lambda_i$ | Eigenvalue of matrix $\Sigma_2^{-1}\Sigma_1$ |
| $\hat{\epsilon}_{\mathrm{f}}$ | Empirical factual error |
| $I(\mathbf{x}; \mathbf{z})$ | Mutual information between input $\mathbf{x}$ and representation $\mathbf{z}$ |
| $\varepsilon_{\mathrm{pehe}}(\mathbf{t}_1, \mathbf{t}_2)$ | Precision in Estimation of Heterogeneous Effect (PEHE) |

## B  Proofs

**Theorem 1** (Counterfactual Generalization Bound, Gaussian Scenario)**.** *Let $\phi$ be an encoder $X \to Z$, and let $f$ be an outcome function $Z \times T \to Y$. Assume that the joint distribution $p(z,t)$ follows a multivariate Gaussian distribution:*

$$p(\mathbf{z}, \mathbf{t}) \sim \mathcal{N}\left( \begin{bmatrix} \mu_z \\ \mu_t \end{bmatrix}, \Sigma_1 \right), \quad where \quad \Sigma_1 = \begin{bmatrix} \Sigma_z & \Sigma_{z,t} \\ \Sigma_{z,t}^T & \Sigma_t \end{bmatrix}.$$

*Let $\Sigma_2$ denote the covariance matrix of the product of marginals $p(\mathbf{z})p(\mathbf{t})$, i.e., the case where $\mathbf{z} \perp \mathbf{t}$. Then,*

$$\Sigma_2 = \begin{bmatrix} \Sigma_z & 0 \\ 0 & \Sigma_t \end{bmatrix}.$$

*Under Assumptions 3 and 4, we have:*

$$\epsilon_{\mathrm{cf}} \leq \epsilon_{\mathrm{f}} + C\sqrt{2 D_{CS}(p_\phi(\mathbf{z}, \mathbf{t}) \,\|\, p_\phi(\mathbf{z})p(\mathbf{t}))}, \tag{29}$$

*if*

$$\sum_{i=1}^{d} \log\left( \frac{2 + \lambda_i + 1/\lambda_i}{4} \right) \geq 4,$$

*where $\lambda_i$ is the $i$-th eigenvalue of*

$$\Sigma_2^{-1}\Sigma_1 = \begin{bmatrix} I & \Sigma_z^{-1}\Sigma_{z,t} \\ \Sigma_t^{-1}\Sigma_{z,t}^T & I \end{bmatrix}.$$

*Proof.* We first introduce Proposition 1 (Yin et al., 2024).

**Proposition 1.** *Let $\Phi$ be the cumulative distribution function of a standard normal distribution. Let $p \sim \mathcal{N}(\mu_1, \Sigma_1)$ and $q \sim \mathcal{N}(\mu_2, \Sigma_2)$ be any $d$-dimensional Gaussian distributions. We have:*

$$D_{\mathrm{TV}} \leq \sqrt{D_{\mathrm{CS}}}, \tag{30}$$

*if one of the following conditions is satisfied:*

- $\Sigma_1 = \Sigma_2 = \Sigma$ and $1/2\sqrt{\delta^\top \Sigma^{-1}\delta} \geq 2\Phi(\|\Sigma^{-1/2}\delta\|_2/2) - 1$, where $\delta = \mu_1 - \mu_2$;

- $\sum_{i=1}^{d} \log\left( \frac{2 + \lambda_i + 1/\lambda_i}{4} \right) \geq 4$, where $\lambda_i$ is the $i$-th eigenvalue of $\Sigma_2^{-1}\Sigma_1$.

*where $D_{TV}$ is the total variation (TV) distance defined as $D_{TV} = \frac{1}{2}\int |p(\mathbf{x}) - q(\mathbf{x})|d\mathbf{x}$.*

By Proposition 1, given two Gaussian distributions

$$p(\mathbf{z}, \mathbf{t}) \sim \mathcal{N}\left( \begin{bmatrix} \mu_z \\ \mu_t \end{bmatrix}, \begin{bmatrix} \Sigma_z & \Sigma_{z,t} \\ \Sigma_{z,t}^T & \Sigma_t \end{bmatrix} \right),$$

and

$$p(\mathbf{z})p(\mathbf{t}) \sim \mathcal{N}\left( \begin{bmatrix} \mu_z \\ \mu_t \end{bmatrix}, \begin{bmatrix} \Sigma_z & 0 \\ 0 & \Sigma_t \end{bmatrix} \right),$$

we have:

$$\frac{1}{2}\int_{\mathcal{T}}\int_{\mathcal{Z}} |p(\mathbf{z}, \mathbf{t}) - p(\mathbf{z})p(\mathbf{t})|d d\mathbf{z}d d\mathbf{t} \leq \sqrt{D_{\mathrm{CS}}(p(\mathbf{z}, \mathbf{t}); p(\mathbf{z})p(\mathbf{t}))}, \tag{31}$$

if

$$\sum_{i=1}^{d_{\mathbf{z}}+d_{\mathbf{t}}} \log\left( \frac{2 + \lambda_i + 1/\lambda_i}{4} \right) \geq 4,$$

where $\lambda_i$ is the $i$-th eigenvalue of

$$\begin{bmatrix} I & \Sigma_z^{-1}\Sigma_{z,t} \\ \Sigma_t^{-1}\Sigma_{z,t}^T & I \end{bmatrix}.$$

We now proceed with the proof of the main theorem.

Let $L$ denote the loss (Definition 4) and $\epsilon_{\text{cf}}$ $\epsilon_{\text{f}}$ the counterfactual and factual losses, respectively. Following the techniques of Kazemi & Ester (2024) and Bellot et al. (2022), we write:

$$
\begin{aligned}
\epsilon_{\text{cf}} - \epsilon_{\text{f}} &= \int_{\mathcal{T}} \epsilon_{\text{f}}^{\ell}(\mathbf{t})p(\mathbf{t})d\mathbf{t} - \int_{\mathcal{T}} \epsilon_{\text{cf}}^{\ell}(\mathbf{t})p(\mathbf{t})d\mathbf{t} \\
&= \int_{\mathcal{X}} \ell_{L,f,\phi}(\mathbf{x},\mathbf{t})\, p(\mathbf{x}|\mathbf{t})\, d\mathbf{x} - \int_{\mathcal{X}} \ell_{L,f,\phi}(\mathbf{x},\mathbf{t})\, p(\mathbf{x})\, d\mathbf{x} \\
&= \int_{\mathcal{T}} \left( \int_{\mathcal{X}} \ell_{L,f,\phi}(\mathbf{x},\mathbf{t}) \Big[ p(\mathbf{x}|\mathbf{t}) - p(\mathbf{x}) \Big] d\mathbf{x} \right) p(\mathbf{t})d\mathbf{t} \\
&= \int_{\mathcal{T}} \left( \int_{\mathcal{X}} \ell_{L,f,\phi}(\mathbf{x},\mathbf{t})\, p(\mathbf{t}) \Big[ p(\mathbf{x}|\mathbf{t}) - p(\mathbf{x}) \Big] d\mathbf{x} \right) d\mathbf{t} \\
&= \int_{\mathcal{T}} \left( \int_{\mathcal{X}} \ell_{L,f,\phi}(\mathbf{x},\mathbf{t}) \Big[ p(\mathbf{t})p(\mathbf{x}|\mathbf{t}) - p(\mathbf{t})p(\mathbf{x}) \Big] d\mathbf{x} \right) d\mathbf{t} \\
&= \int_{\mathcal{T}} \left( \int_{\mathcal{X}} \ell_{L,f,\phi}(\mathbf{x},\mathbf{t}) \Big[ p(\mathbf{x},\mathbf{t}) - p(\mathbf{t})p(\mathbf{x}) \Big] d\mathbf{x} \right) d\mathbf{t} \\
&= \int_{\mathcal{T}} \int_{\mathcal{X}} \ell_{L,f,\phi}(\mathbf{x},\mathbf{t}) \Big[ p(\mathbf{x},\mathbf{t}) - p(\mathbf{t})p(\mathbf{x}) \Big] d\mathbf{x}\, d\mathbf{t}.
\end{aligned}
\tag{32}
$$

Following Kazemi & Ester (2024) and Bellot et al. (2022), we can then apply a change of variables.

$$
\begin{aligned}
\epsilon_{\text{cf}} - \epsilon_{\text{f}} &= \int_{\mathcal{T}} \int_{\mathcal{X}} \ell_{L,f,\phi}(\mathbf{x},\mathbf{t}) \Big[ p(\mathbf{x},\mathbf{t}) - p(\mathbf{x})p(\mathbf{t}) \Big] d\mathbf{x}\, d\mathbf{t} \\
&= \int_{\mathcal{T}} \int_{\mathcal{Z}} \ell_{L,f,\phi}(\phi(\mathbf{z}),\mathbf{t}) \Big[ p(\phi(\mathbf{z}),\mathbf{t}) - p(\phi(\mathbf{z}))p(\mathbf{t}) \Big] d\mathbf{z}\, d\mathbf{t} \\
&= \int_{\mathcal{T}} \int_{\mathcal{Z}} \ell_{L,f,\phi}(\phi(\mathbf{z}),\mathbf{t}) \Big[ p(\phi(\mathbf{z}),\mathbf{t}) - p(\phi(\mathbf{z}))p(\mathbf{t}) \Big] \left| J_\phi J_\phi^{-1} \right| d\mathbf{z}\, d\mathbf{t} \\
&= \int_{\mathcal{T}} \int_{\mathcal{Z}} \ell_{L,f,\phi}(\phi(\mathbf{z}),\mathbf{t}) \Big[ p_\phi(\mathbf{z},\mathbf{t}) - p_\phi(\mathbf{z})p(\mathbf{t}) \Big] d\mathbf{z}\, d\mathbf{t} \\
&\leq \int_{\mathcal{T}} \int_{\mathcal{Z}} C \Big| p_\phi(\mathbf{z},\mathbf{t}) - p_\phi(\mathbf{z})p(\mathbf{t}) \Big|
\end{aligned}
\tag{33}
$$

Combine Eq. (31) with Eq. (33), we have:

$$
\epsilon_{\text{cf}} \leq \epsilon_{\text{f}} + C\, \sqrt{2\, D_{CS}(p(\mathbf{x},\mathbf{t}) \| p(\mathbf{x})p(\mathbf{t}))}.
\tag{34}
$$

$\square$

**Proposition 1.** *Let $\phi$ be an encoder mapping $X \to Z$, and let $f$ be an outcome function $Z \times T \to Y$. Assume that $p(\mathbf{z}, \mathbf{t})$ is an arbitrary joint distribution. Then, we have*

$$\epsilon_{\mathrm{cf}} \lesssim \epsilon_{\mathrm{f}} + C\sqrt{2D_{CS}(p_\phi(\mathbf{z}, \mathbf{t}) \,\|\, p_\phi(\mathbf{z})p(\mathbf{t}))},$$

*where $\lesssim$ denotes "less than or approximately equal to," and the precise conditions under which this inequality holds are discussed in Appendix B.*

*Proof.* The logic of the proof of Proposition 1 follows that of Theorem 1, but relies on a more general relationship between total variation and Cauchy-Schwarz divergence, as stated in Proposition 2.

**Proposition 2.** *(Yin et al., 2024) For any density functions $p$ and $q$, and any $\epsilon > 0$, let $\mathcal{A}_\epsilon = \{\mathbf{x} : p(\mathbf{x}) \leq \epsilon\} \cup \{\mathbf{x} : q(\mathbf{x}) \leq \epsilon\}$ and $\mathcal{A}_\epsilon^{\complement}$ be its complement (see Fig. 4). Moreover, define $T_{\mathcal{A}_\epsilon^{\complement}} = \sup\left\{p(\mathbf{x})q(\mathbf{x}), \mathbf{x} \in \mathcal{A}_\epsilon^{\complement}\right\}$ and $\left|\mathcal{A}_\epsilon^{\complement}\right|$ to denote the "length" of the set $\mathcal{A}_\epsilon^{\complement}$ (strictly speaking, the Lebesgue measure of the set $\mathcal{A}_\epsilon^{\complement}$). Suppose there exists an $\epsilon > 0$ such that $T_{\mathcal{A}_\epsilon^{\complement}}\left|\mathcal{A}_\epsilon^{\complement}\right| < \infty$ and $C = \int p^2(\mathbf{x})\,d\mathbf{x} \int q^2(\mathbf{x})\,d\mathbf{x} \geq \exp(2)\left(2\epsilon + T_{\mathcal{A}_\epsilon^{\complement}}\left|\mathcal{A}_\epsilon^{\complement}\right|\right)^2$, then*

$$D_{\mathrm{TV}}(p;q) \leq \sqrt{D_{\mathrm{CS}}(p;q)}. \tag{35}$$

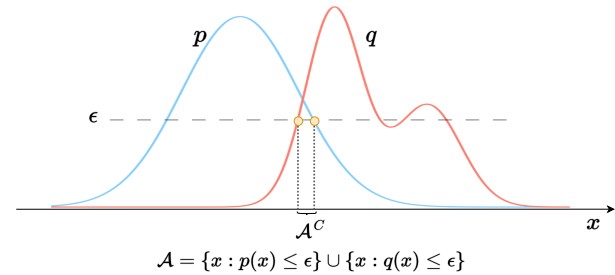

$$\mathcal{A} = \{x : p(x) \leq \epsilon\} \cup \{x : q(x) \leq \epsilon\}$$

Figure 4: A graphical illustration of the sets $\mathcal{A}_\epsilon$ and $\mathcal{A}_\epsilon^{\complement}$ defined in Proposition 2.

In our context, it is difficult to ensure that $p = p(\mathbf{z}, \mathbf{t})$ and $q = p(\mathbf{z})p(\mathbf{t})$ do not significantly overlap. That is, although we theoretically aim to enforce $p = q$, achieving perfect independence between $\mathbf{z}$ and $\mathbf{t}$ is rarely feasible in practice. This is because eliminating all statistical dependence between $\mathbf{z}$ and $\mathbf{t}$ requires removing all treatment-predictive signals from $\mathbf{x}$, which is often incompatible with preserving outcome-relevant information.

Therefore, the condition $\int p^2(\mathbf{x})\,d\mathbf{x} \int q^2(\mathbf{x})\,d\mathbf{x} \geq \exp(2)\left(2\epsilon + T_{\mathcal{A}_\epsilon^{\complement}}\left|\mathcal{A}_\epsilon^{\complement}\right|\right)^2$ for some $\epsilon > 0$ is largely satisfied, then $D_{\mathrm{TV}} \leq \sqrt{D_{\mathrm{CS}}}$.

$\square$

**Proposition 3** (PEHE Error Bound). *Given an encoder $\phi$ and outcome prediction function $f$ and a unit-loss function $\ell_{L,f,\phi}(\mathbf{x}, \mathbf{t})$ that satisfies Assumption 4 and its associated $L$ is squared error $\|\cdot\|^2$, the following inequality holds:*

$$
\begin{aligned}
\varepsilon_{pehe}(\mathbf{t}_1, \mathbf{t}_2) \leq \ & \varepsilon_{\mathrm{f}}^{\ell}(\mathbf{t}_1) + \varepsilon_{\mathrm{f}}^{\ell}(\mathbf{t}_2) \\
& + \sqrt{2 D_{CS}\left(p_{\phi}(\mathbf{z}) \| p_{\phi}(\mathbf{z}|\mathbf{t}_1)\right)} + \sqrt{2 D_{CS}\left(p_{\phi}(\mathbf{z}) \| p_{\phi}(\mathbf{z}|\mathbf{t}_2)\right)}.
\end{aligned}
\tag{36}
$$

*Proof.* Recall the definition of the factual error:

$$
\epsilon_{\mathrm{f}}^{\ell}(\mathbf{t}) = \int_{\mathcal{X}} \ell_{L,f,\phi}(\mathbf{x}, \mathbf{t}) \, p(\mathbf{x}|\mathbf{t}) \, d\mathbf{x}
\tag{37}
$$

and the counterfactual error

$$
\epsilon_{\mathrm{cf}}^{\ell}(\mathbf{t}) = \int_{\mathcal{T}'=[0,1]\backslash\{\mathbf{t}\}} \int_{\mathcal{X}} \ell_{L,f,\phi}(\mathbf{x}, \mathbf{t}) \, p(\mathbf{x}|\mathbf{t}') \, d\mathbf{x} \, d\mathbf{t}'.
\tag{38}
$$

Also recall the definition of the expected precision of estimating heterogeneous effect (PEHE) between treatment pairs $\mathbf{t}_1, \mathbf{t}_2 \in \mathcal{T}$:

$$
\varepsilon_{\mathrm{pehe}}(\mathbf{t}_1, \mathbf{t}_2) = \int_{\mathcal{X}} \Big[ (\mu(\mathbf{x}, \mathbf{t}_1) - \mu(\mathbf{x}, \mathbf{t}_2)) - (f(\phi(\mathbf{x}), \mathbf{t}_1) - f(\phi(\mathbf{x}), \mathbf{t}_2)) \Big]^2 p(\mathbf{x}) \, d\mathbf{x}.
\tag{39}
$$

Now, following existing binary generalization bounds (e.g. Shalit et al. (2017)), we know that the PEHE can be bounded by twice the factual error and twice the counterfactual error (minus a term that accounts for the effective variance of the outcome distribution $\sigma_Y^2$). We have

$$
\varepsilon_{\mathrm{pehe}}(\mathbf{t}_1, \mathbf{t}_2) \leq \varepsilon_{\mathrm{f}}^{\ell}(\mathbf{t}_1) + \varepsilon_{\mathrm{f}}^{\ell}(\mathbf{t}_2) + \varepsilon_{\mathrm{cf}}^{\ell}(\mathbf{t}_1) + \varepsilon_{\mathrm{cf}}^{\ell}(\mathbf{t}_2) - \sigma_Y^2.
\tag{40}
$$

Since, by Proposition 1, our counterfactual error is bounded by

$$
C \sqrt{2 \, D_{CS}\Big(p_{\phi}(\mathbf{z}, \mathbf{t}) \| p_{\phi}(\mathbf{z}) p(\mathbf{t})\Big)},
\tag{41}
$$

we have that (since we no longer integrate over $\mathbf{t}$) for each of the terms

$$
\sqrt{2 \, D_{CS}\Big(p_{\phi}(\mathbf{z}) \| p_{\phi}(\mathbf{z}) p(\mathbf{t}_i)\Big)},
\tag{42}
$$

where $i$ denotes the treatment regimen. As this term upper bounds the counterfactual error (as per Proposition 1), it immediately follows that

$$
\begin{aligned}
\varepsilon_{\mathrm{pehe}}(\mathbf{t}_1, \mathbf{t}_2) \leq \ & \varepsilon_{\mathrm{f}}^{\ell}(\mathbf{t}_1) + \varepsilon_{\mathrm{f}}^{\ell}(\mathbf{t}_2) \\
& + \sqrt{2 D_{\mathrm{CS}}\left(p_{\phi}(\mathbf{z}) \| p_{\phi}(\mathbf{z}|\mathbf{t}_1)\right)} + \sqrt{2 D_{\mathrm{CS}}\left(p_{\phi}(\mathbf{z}) \| p_{\phi}(\mathbf{z}|\mathbf{t}_2)\right)}.
\end{aligned}
\tag{43}
$$

$\square$

# C  Algorithm

---

**Algorithm 1** IBEX - Training Algorithm

---

**Require:** Dataset $\mathcal{D} = \{(\mathbf{x}_i, \mathbf{t}_i, y_i)\}_{i=1}^N$; epochs $T$; batch size $B$; learning rate $\eta$;
1: hyper-parameters $\beta, \gamma, \lambda$; kernel function $\kappa(\cdot, \cdot)$
**Ensure:** Trained parameters $\theta_\phi, \theta_\tau, \theta_f$
2: Initialize $\theta_\phi, \theta_\tau, \theta_f$ (e.g. via AdamW optimizer)
3: **for** epoch = 1 to $T$ **do**
4:     Shuffle $\mathcal{D}$ and split into $\lceil N/B \rceil$ batches
5:     **for all** mini-batch $\mathcal{B} = \{(\mathbf{x}_j, \mathbf{t}_j, y_j)\}_{j=1}^B$ **do**
6:         Encode covariates: $\mathbf{z}_j \leftarrow \phi_{\theta_\phi}(\mathbf{x}_j)$
7:         Encode treatments: $\tilde{\mathbf{t}}_j \leftarrow \tau_{\theta_\tau}(\mathbf{t}_j)$
8:         Predict outcome: $\hat{y}_j \leftarrow f_{\theta_f}(\mathbf{z}_j, \tilde{\mathbf{t}}_j)$
9:         Compute prediction loss:

$$\mathcal{L}_{\text{pred}} \leftarrow \frac{1}{B} \sum_{j=1}^B (y_j - \hat{y}_j)^2$$

10:        Compute Gram matrices:

$$K_{ij} := \kappa(\mathbf{z}_i, \mathbf{z}_j), \quad Q_{ij} := \kappa(\mathbf{t}_i, \mathbf{t}_j)$$

11:        Compute Cauchy-Schwarz divergence loss:

$$\mathcal{L}_{\text{CS}} \leftarrow \log\left(\frac{\text{tr}(KQ)}{B^2}\right) + \log\left(\frac{\mathbf{1}^\top K \mathbf{1} \cdot \mathbf{1}^\top Q \mathbf{1}}{B^4}\right) - 2\log\left(\frac{\mathbf{1}^\top KQ\mathbf{1}}{B^3}\right)$$

12:        Compute sample covariance:

$$\bar{\mathbf{z}} \leftarrow \frac{1}{B} \sum_{j=1}^B \mathbf{z}_j, \quad \Sigma_z \leftarrow \frac{1}{B} \sum_{j=1}^B (\mathbf{z}_j - \bar{\mathbf{z}})(\mathbf{z}_j - \bar{\mathbf{z}})^\top$$

13:        Compute dimensionality penalty:

$$\mathcal{L}_{\text{dim}} \leftarrow \|Z^\top\|_{2,1} + \lambda \cdot \log\det(\Sigma_z + \varepsilon I)$$

14:        Combine total loss:

$$\mathcal{L}_{\text{IBEX}} \leftarrow \mathcal{L}_{\text{pred}} + \beta \cdot \mathcal{L}_{\text{dim}} + \gamma \cdot \mathcal{L}_{\text{CS}}$$

15:        Update parameters $\theta_\phi, \theta_\tau, \theta_f$ using gradient descent:

$$\theta \leftarrow \theta - \eta \cdot \nabla_\theta \mathcal{L}_{\text{IBEX}}$$

16:     **end for**
17: **end for**
18: **return** $\theta_\phi, \theta_\tau, \theta_f$

---

# D   Additional Analyses & Experiments

**Empirical Comparison of Hölder and KL Divergence Bounds**

To visually compare the tightness of our proposed counterfactual generalization bound based on the Cauchy-Schwarz (CS) divergence with existing bounds that rely on KL divergence and Pinsker's inequality, we conducted a synthetic numerical experiment. We considered a target joint distribution $p(z,t) = \mathcal{N}([0,0], I)$, representing the observational distribution in latent-outcome space. The reference distribution $p(z)p(t)$ was constructed as a product of marginals, where $p(t) = \mathcal{N}(0,1)$ remained fixed and $p(z) = \mathcal{N}(\mu, 1)$ was shifted across a range of means $\mu \in [-2, 2]$. For each shift, we evaluated both the KL divergence $D_{\mathrm{KL}}(p(z,t) \,\|\, p(z)p(t))$ and the Hölder divergence $D_H$, defined via generalized Hölder's inequality with exponent $a = b = 2$, corresponding to the Cauchy-Schwarz divergence. Both divergences were computed numerically over a discretized grid.

The results, plotted in Figure 5, show that both divergences increase symmetrically with the shift in $p(z)$, as expected. The CS divergence grows more slowly than the KL divergence, reflecting a tighter measure of distributional shift. This behavior supports our theoretical result that bounds using Hölder-type divergences can yield sharper generalization guarantees than those relying on KL divergence, particularly when the distributions are close. These findings suggest that CS-based generalization bounds not only have solid theoretical underpinnings but also offer empirical advantages in scenarios where covariate shift is moderate.

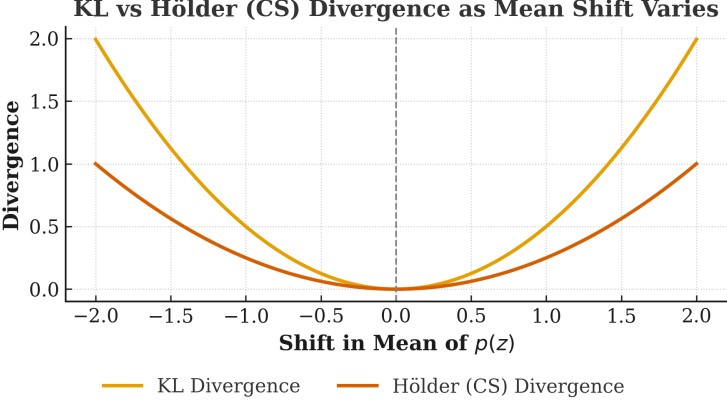

Figure 5: Comparative Example of the Kullback-Leibler with the Cauchy-Schwarz bound in an idealized setting.

**Ablation Experiment on Regularization Sensitivity**

In this experiment, we examine the effect of the two main regularizers ($\beta$ and $\gamma$) on the two counterfactual metrics. We also investigate the impact of the "internal" hyperparameter used in the log-expression of $R_{\dim}(\mathbf{z})$. Figure 6 illustrates the influence of the two primary hyperparameters of the IBEX model, as measured by $\sqrt{\text{MISE}}$ and $\sqrt{\text{PE}}$, on the MIMIC-IV dataset, with weight values ranging from 0 to 0.3.

In the setting where we test the effect of $\beta$ (Figure 6, left panel), we vary only the $\beta$-hyperparameter while fixing $\gamma = 0$. In the $\gamma$-test setting (right panel), we reverse this: $\beta$ is fixed while $\gamma$ is varied. To conserve computational resources, we train the model for only 1000 epochs in this ablation study, compared to at least 3000 epochs in the main experiment.

The general trend is that the regularizers have a significant impact on $\sqrt{\text{PE}}$, whereas $\sqrt{\text{MISE}}$ tends to be more robust with respect to $\beta$, which controls the dimensionality of $\mathbf{z}$. However, $\sqrt{\text{MISE}}$ does show some sensitivity to $\gamma$, which governs the strength of the Cauchy–Schwarz regularization term. This is theoretically expected, as this regularizer more directly influences prediction performance.

Figure 7 shows the effect of the $\lambda$ parameter on the MIMIC-IV dataset. We keep the other two main parameters fixed ($\gamma = 0.01$ and $\beta = 0.1$), consistent with the main experiment, and train the model for only 1000 epochs. As shown, there is essentially no effect on the two counterfactual metrics. However, for the out-of-distribution loss, we observe a small difference.

We hypothesize that this limited impact is due to the diminishing influence of the $\lambda$-term relative to the primary prediction objective, the additional contribution from the Cauchy–Schwarz regularizer, and the fact that the product of small values (e.g., $\lambda\beta = 0.05 \times 0.01 = 0.0005$) results in a negligible effect. Despite the modest influence, we retain the term in our implementation due to its favorable theoretical properties and its slight positive contribution to the loss.

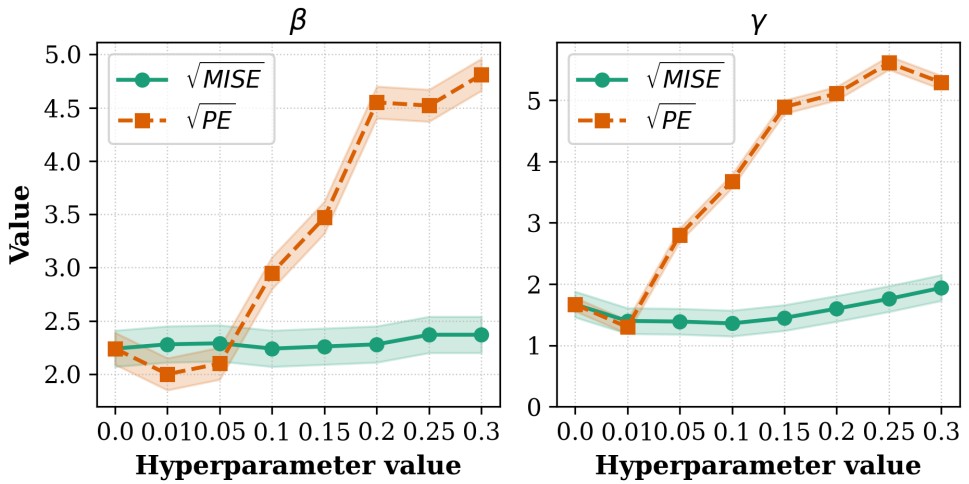

Figure 6: Effect of the two main regularizers, $\beta$ and $\gamma$, on counterfactual performance metrics for the MIMIC-IV dataset. The weight values range from 0 to 0.3.

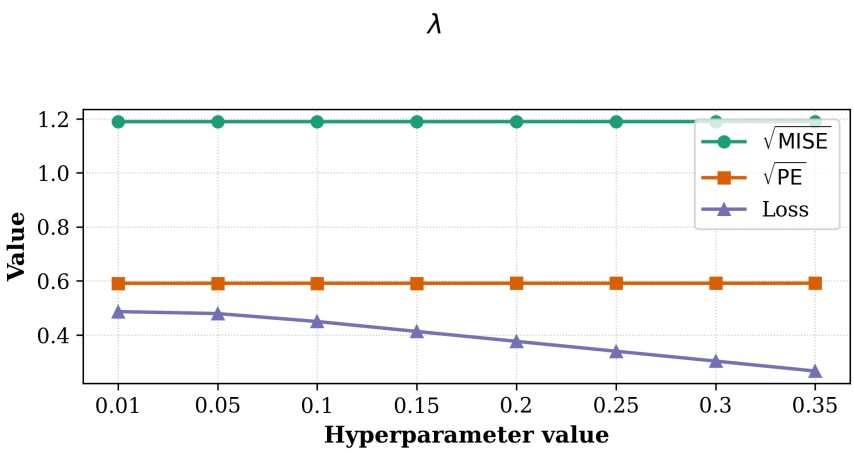

Figure 7: Effect of the hyperparameter regularizer $\lambda$ on the MIMIC-IV dataset. Performance is evaluated using counterfactual metrics and out-of-distribution loss.

**Overlap Stress-test**

To assess robustness to violations of overlap in continuous-treatment settings, we conduct an explicit support-mismatch stress test using semi-synthetic outcomes constructed on real TCGA covariates. As before, treatments are generated according to the data-generating process, but the training set is restricted to units whose observed treatment lies within a truncated interval $t \in [0.2, 0.8]$, while the test set and evaluation dose grid span the full range $t \in [0, 1]$. This induces a deliberate mismatch between the treatment support observed during training and the target support at evaluation, forcing models to extrapolate counterfactual outcomes beyond the observed region. Performance is evaluated using the square root of the mean integrated squared error (MISE), reported separately for doses inside and outside the training support, along with the policy error computed over the full dose grid. This experimental setting directly probes sensitivity to weak overlap, which is known to pose a fundamental challenge in continuous-treatment causal inference.

| Method | TCGA (Full) | TCGA (In-Support) | TCGA (Out-of-Support) |
|--------|-------------|-------------------|------------------------|
| SCIGAN | **0.56 ± 0.26** | 0.36 ± 0.12 | **0.75 ± 0.42** |
| IBEX | 0.70 ± 0.34 | **0.20 ± 0.11** | 1.06 ± 0.58 |
| ACFR | 1.22 ± 0.82 | 0.84 ± 0.52 | 1.63 ± 1.13 |

Table 4: Out-of-sample performance under overlap stress test on TCGA, measured by $\sqrt{\text{MISE}}$ (lower is better). Training is restricted to treatments $t \in [0.2, 0.8]$. *In-Support* and *Out-of-Support* errors are computed on dose grid points inside and outside the training support, respectively. Best performance within each column is shown in bold.

Table 4 compares SCIGAN, IBEX, and ACFR under an overlap stress test on TCGA, where training is restricted to treatments in $[0.2, 0.8]$. SCIGAN achieves the best performance on the full evaluation grid as well as on out-of-support doses, indicating stronger extrapolation capability when predicting treatment effects outside the observed support. In contrast, IBEX substantially outperforms the other methods on in-support doses, achieving the lowest error within the training range, but exhibits noticeably higher error and variance when evaluated outside the training support. ACFR performs consistently worse across all settings, with particularly large errors and variability on out-of-support doses, suggesting limited robustness to overlap violations. Overall, these results highlight a clear trade-off between interpolation and extrapolation: IBEX is most effective when sufficient treatment overlap is present, whereas SCIGAN is more robust to overlap violations and generalizes better to unseen treatment regions.

**Synthetic comparison of CS and KL bounds.**

We construct a synthetic factual dataset $\mathcal{D}_f$ by sampling covariates $\mathbf{x} \in \mathcal{X}$ and generating continuous treatments $\mathbf{t}_f$ such that a scalar parameter $\alpha$ controls the strength of dependence between $\mathbf{x}$ and $\mathbf{t}_f$. Outcomes are generated from a nonlinear dose-response function $\mu(\mathbf{t}, \mathbf{x})$ with heterogeneous treatment effects. For each value of $\alpha$, we train an outcome model $f(\mathbf{z}, \tilde{\mathbf{t}})$ on factual data and evaluate the expected counterfactual error $\epsilon_{cf}$ using the mean integrated squared error over a grid of treatments.

To isolate the effect of the divergence term in the theoretical bound, we use the identity encoder $\phi(\mathbf{x}) = \mathbf{z} = \mathbf{x}$ and estimate the dependence between $\mathbf{z}$ and $\mathbf{t}$ using both the Cauchy-Schwarz divergence $D_{CS}(p(\mathbf{z}, \mathbf{t}) \| p(\mathbf{z})p(\mathbf{t}))$ and the Kullback–Leibler divergence $D_{KL}(p(\mathbf{z}, \mathbf{t}) \| p(\mathbf{z})p(\mathbf{t}))$. We construct empirical proxy bounds of the form $\hat{\epsilon}_f + c\sqrt{D(\mathbf{z}; \mathbf{t})}$, where $\hat{\epsilon}_f$ is the empirical factual error and the same constant $c$ is shared across divergences. A smaller proxy indicates a tighter bound.

Figure 8 provides a simple synthetic illustration of the relationship between the theoretical bound and counterfactual error. As the confounding strength $\alpha$ increases, the empirical counterfactual error $\epsilon_{cf}$ increases, reflecting the growing difficulty of extrapolation from factual data. Both divergence-based proxies track this trend through the dependence between $\mathbf{z}$ and $\mathbf{t}$, but the proxy constructed using the Cauchy–Schwarz divergence remains uniformly smaller than the corresponding KL-based proxy under the same scaling.

Importantly, this experiment is not intended to suggest that a tighter theoretical bound yields better empirical performance. The learned predictor and training procedure are identical across divergences, and the counterfactual error itself does not depend on the choice of divergence. Rather, the figure illustrates that, under increasing confounding, the CS-based proxy provides a strictly tighter upper control on counterfactual error than the KL-based proxy while preserving the same qualitative dependence on $\alpha$. This supports the theoretical motivation for using Cauchy–Schwarz divergence as a sharper analytical tool, while making explicit that bound tightness and empirical accuracy are conceptually distinct.

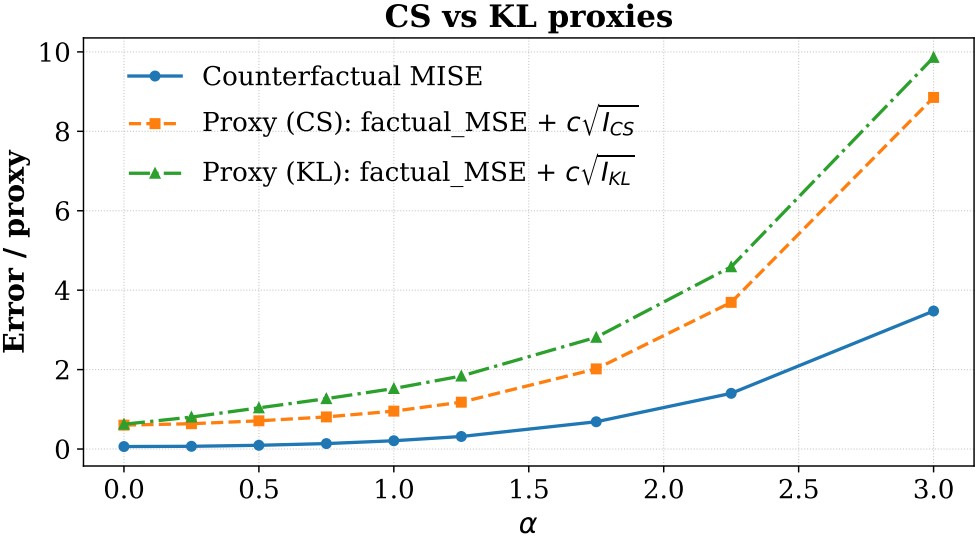

Figure 8: Counterfactual error and divergence-based proxy criteria across confounding regimes. The horizontal axis shows the confounding strength $\alpha$. The vertical axis reports the empirical counterfactual error (MISE) of the outcome predictor $f(\mathbf{z}, \tilde{\mathbf{t}})$ and two proxy quantities of the form $\hat{\epsilon}_f + \sqrt{D(p_{\mathbf{z},\mathbf{t}} \| p_{\mathbf{z}}p_{\mathbf{t}})}$. Here $\hat{\epsilon}_f$ denotes the empirical factual error on $\mathcal{D}_f$, $\mathbf{z} = \phi(\mathbf{x})$ is the covariate representation (in this experiment $\phi$ is the identity so $\mathbf{z} = \mathbf{x}$), and $D(\cdot \| \cdot)$ is instantiated as either the Cauchy–Schwarz divergence $D_{CS}$ or the Kullback–Leibler divergence $D_{KL}$.

# E    Dataset Descriptions

### E.1    MIMIC-IV

The MIMIC-IV database (Johnson et al., 2023) contains detailed, de-identified health records from patients admitted to intensive care units, making it a valuable resource for studying treatment strategies in critical care. In this work, we focus on estimating continuous treatment effects related to mechanical ventilation. The treatment variable $T$ represents a continuous configuration of ventilator settings—specifically, tidal volume and respiratory rate—both of which are critical for patient outcomes.

Following largely the processing steps of Schwab et al. (2020), we filter MIMIC-IV to include only patients who received mechanical ventilation, yielding a cohort of 5,476 individuals. Each patient is represented by a vector of 33 covariates $X$, including demographics (e.g., age, sex) and clinical measurements (e.g., respiratory indicators, cardiac function). Unlike conventional approaches that discretize $T$ (e.g., into dosage bins), we model it as a continuous vector to capture the joint influence of multiple ventilator parameters. The dataset is publicly available at: `https://physionet.org/content/mimiciv/2.2/`. We follow the treatment assignment and outcome-generation mechanisms of Schwab et al. (2020) and Kazemi & Ester (2024). In this experiment, our treatment is two-dimensional.

### E.2    News

The News dataset (Asuncion et al., 2007) comprises bag-of-words representations of randomly sampled *New York Times* articles. It contains a 3,477-dimensional word count vector. Following prior work (Bica et al., 2020), we use a subset of 5,000 samples and synthetically generate continuous treatment and outcome variables to evaluate model performance. Moreover, following Schwab et al. (2020) and Bica et al. (2020), we train an initial topics model $q$ over the covariates $q(\mathbf{x})$. The dataset is available at: `https://archive.ics.uci.edu/dataset/164/bag+of+words`. We use the same treatment mechanism as in Kazemi & Ester (2024).

### E.3    TCGA

The TCGA (The Cancer Genome Atlas Program) Weinstein et al. (2013) dataset contains gene expression data for cancer patients. We selected a total of 9000 patients and the 4000 most variable genes. The gene expression values were and scaled to fall within the $[0, 1]$ range. Additionally, features were normalized to have unit norm for each patient. The outcome can be understood as how likely cancer recurrence is. The dataset version used is the same as the one employed in DRNet, available at: `https://github.com/d909b/drnet`.

## F  Assignment Mechanisms

**Semi-synthetic data generation.**  We follow Bica et al. (2020) and Kazemi & Ester (2024) to simulate outcomes using real-world covariates $\mathbf{x}$ sourced from publicly available datasets—specifically, TCGA, News, and MIMIC-III. For MIMIC-IV specifically, we imitate the bivariate treatment assignment of Schwab et al. (2020) (they use it for an older version of MIMIC, MIMIC-III). Our setup involves three treatments, each associated with a continuous dosage. For every treatment $w$, we sample three latent parameter vectors $\mathbf{v}_1^w$, $\mathbf{v}_2^w$, and $\mathbf{v}_3^w$. These are generated by drawing raw vectors $\mathbf{u}_i^w \sim \mathcal{N}(0,1)$ and normalizing them: $\mathbf{v}_i^w = \mathbf{u}_i^w / \|\mathbf{u}_i^w\|$ (using the Euclidean norm).

Each treatment's response function $f_w(\mathbf{x}, d)$, which depends on the covariates and dosage, is described in Table 5, including the analytically derived optimal dosage $d_w^*$. Gaussian noise $\epsilon \sim \mathcal{N}(0, 0.2)$ is added to simulate observational variability.

**Intervention assignment**  To assign treatments, we begin by sampling dosages $d_w$ for each treatment from a Beta distribution parameterized by:

$$d_w \mid \mathbf{x} \sim \text{Beta}(\alpha, \beta_w), \quad \alpha \geq 1,$$

where $\alpha$ governs the extent of dosage bias. $\alpha = 1$ yields a uniform distribution, while higher values induce peaked distributions. The parameter $\beta_w$ is calibrated as:

$$\beta_w = \frac{\alpha - 1}{d_w^*} + 2 - \alpha,$$

which ensures that the mode of the Beta distribution aligns with the treatment's optimal dosage $d_w^*$. Given the sampled dosages, we determine the treatment assignment via:

$$w_f \mid \mathbf{x} \sim \text{Categorical}(\text{softmax}(\kappa f(\mathbf{x}, d_w)))$$

where the temperature parameter $\kappa$ adjusts the strength of selection bias—larger $\kappa$ values favor the highest response, while $\kappa = 0$ results in random treatment assignment. The final factual treatment and dosage pair is denoted by $(w_f, d_{w_f})$. Unless otherwise specified, we set $\alpha = 2$ and $\kappa = 2$. We set $C = 10$.

| Dataset | #Samples | #Covariates | Outcome function(s) $f_w(\mathbf{x}, d)$ |
|---|---|---|---|
| TCGA | 9659 | 4000 | $f_w(\mathbf{x}, d) = C(\mathbf{v}_1^{w\top}\mathbf{x} + 12\,\mathbf{v}_2^{w\top}\mathbf{x}\,d - 12\,\mathbf{v}_3^{w\top}\mathbf{x}\,d^2)$ |
| News | 5000 | 3477 | $f_w(\mathbf{x}, d) = C(\mathbf{v}_1^{w\top}\mathbf{x} + \sin\left(\frac{\mathbf{v}_2^{w\top}\mathbf{x}}{\mathbf{v}_3^{w\top}\mathbf{x}}\pi d\right))$ |
| MIMIC-IV | 5476 | 33 | $f_1(\mathbf{x}, d) = C(\mathbf{v}_1^{1\top}\mathbf{x} + 12\,\mathbf{v}_2^{1\top}\mathbf{x}\,d - 12\,\mathbf{v}_3^{1\top}\mathbf{x}\,d^2), \quad f_2(\mathbf{x}, d) = C(\mathbf{v}_1^{2\top}\mathbf{x} + \sin\left(\frac{\mathbf{v}_2^{2\top}\mathbf{x}}{\mathbf{v}_3^{2\top}\mathbf{x}}\pi d\right))$ |

Table 5: Dataset specifications and outcome-generating functions for each treatment $w$. Covariates $\mathbf{x}$ are real, $C = 10$ is a global scaling constant, and $\mathbf{v}_i^w$ are normalized latent vectors.

# G   Implementation Details of Baseline Methods

In this section, we briefly outline key considerations related to the implementation of baseline models. While we generally used the same hyperparameter values as reported in the original works, we occasionally made adjustments to components of the codebases to better align them with our experimental setup.

- **SCIGAN**
    - Source code: `https://github.com/ioanabica/SCIGAN`.
    - We were able to run the original implementation without any major modifications.
    - The method integrated well with our experimental pipeline with minimal adaptation required.

- **DRNet (Wasserstein + HSIC)**
    - Source code: `https://github.com/d909b/drnet`, the official implementation by Schwab et al. (2020).
    - No major structural changes were necessary; only minor adjustments to data formatting were made.
    - We used custom implementation of the Wasserstein metric and the HSIC regularizer from Nagalapatti et al. (2024).

- **VCNet (Wasserstein + HSIC)**
    - We implemented the base VCNet architecture ourselves, as IBEX is fundamentally based on VCNet.
    - To facilitate experimentation, we added configuration switches to:
        * Enable/disable attention mechanisms.
        * Enable/disable spline-based transformations.
    - We incorporated:
        * The HSIC regularizer from Nagalapatti et al. (2024).
        * A custom implementation of the Wasserstein metric.

- **GPS**
    - Implemented from scratch using `scikit-learn`'s machine learning modules Pedregosa et al. (2011).

- **MLP (Multi-Layer Perceptron)**
    - Implemented from scratch using `scikit-learn`'s MLP module.

- **ACFR**
    - Source code: `https://github.com/amirrezakazemi/acfr/tree/main`
    - We slightly modified the original implementation to ensure compatibility with our environment.
    - All core architectural and learning components were preserved.

- **GIKS**
    - Source code: `https://github.com/nlokeshiisc/GIKS_release`. We re-implemented most components of the original codebase to ensure integration with our training pipeline.
    - The regularization terms were preserved from the original implementation.
    - We embedded the GIKS regularizers into our custom VCNet-style architecture, a choice motivated by:
        * **Fair comparison**: using a shared base model improves comparability across methods.
        * **Practical reasons**: integrating the original code proved technically challenging in our setup.

# H   Model Hyperparameters

Table 6: Hyperparameters used for IBEX-CS_Counterfactual_Net

| Component | Value |
|---|---|
| *Network Architecture* | |
| Latent treatment dimension ($t_{\text{dim\_latent}}$) | 8 |
| Latent representation dimension ($z_{\text{dim}}$) | 128 |
| Outcome dimension ($y_{\text{dim}}$) | 1 |
| Hidden dimension (hidden_dim) | 512 |
| Hidden dim for treatment encoder (hidden_dim_t) | 8 |
| Attention dimension (attn_dim) | 64 |
| *Spline Encoder* | |
| Number of B-spline basis functions | 10 |
| Spline degree | 3 |
| Treatment range | [0.0, 1.0] |
| *Optimization* | |
| Optimizer | AdamW |
| Learning rate | 1e-4 |
| Weight decay | 1e-4 |
| Loss function | MSELoss |
| Epochs | 3000 |
| *Regularization and Loss Coefficients* (standard) | |
| $\beta$ (i.e. Cauchy metric regularizer) | 0.1 |
| $\gamma$ (i.e. weight for latent space dim. regularizer) | 0.001 |
| $\lambda$ (secondary hyperparameter to dim.-regularizer) | 0.05 |

# I   Source Code

Our source code can be found at `https://github.com/ljsmalbil/IBEX` and provides a runnable script for the News data set. The other two data sets (TCGA and MIMIC-IV) require additional certification and cannot be shared by third parties (us). However, anyone interested in reproducing the results can acquire the data themselves fairly easily and use our data processing scripts provided in the code base.

