# OpenReview forum: "Continuous Treatment Effect Estimation with Cauchy-Schwarz Divergence Information Bottleneck"
_TMLR — Accepted by TMLR_

### Review · Reviewer_sMnY · 2026-01-05

**Summary Of Contributions:**

This paper proposes IBEX, a representation-learning method for estimating conditional dose–response functions under continuous and multivariate treatments. The main technical claims are: (i) a counterfactual error generalization bound using Cauchy–Schwarz (CS) divergence (argued to be tighter than KL-based variants), and (ii) strengthening via an information-bottleneck (IB) style compression argument to improve generalization. Methodologically, IBEX combines an outcome prediction loss with a CS-divergence-based dependence penalty encouraging treatment-invariant representations and a capacity/compression regularizer on the latent representation. Empirically, the method is evaluated on three benchmarks (MIMIC-IV, TCGA, News) and reports strong performance and robustness to treatment-assignment bias.

**Strengths:** clear motivation for continuous/multivariate treatments, good empirical results, principled link to prior representation-balancing approaches.

**Weaknesses:** several theoretical assumptions/claims appear misaligned with the implemented model (notably around bijectivity vs compression), the "CS tighter than KL" bounds claim is not supported as stated, some experimental reporting/analysis details reduce confidence.

**Audience:**

Yes

**Audience Explanation:**

Continuous and multivariate treatment effect estimation is an important and under-served area in causal inference. The paper will be of interest to the TMLR audience because it adapts recent Cauchy–Schwarz–divergence–based information bottleneck techniques developed for regression (e.g., Yu et al. (2024)) to causal representation learning, aiming to improve upon the KL-based dependence terms used in prior generalization bounds. The proposed kernelized CS regularization and accompanying architecture are evaluated on standard benchmarks (MIMIC-IV, TCGA, News) against strong baselines, further supporting the paper’s relevance, even though some theoretical claims require clarification and tightening.

**Broader Impact Concerns:**

No major broader-impact concerns beyond standard issues in observational causal inference which the authors already address.

**Claims And Evidence:**

No

**Claims Explanation:**

The empirical results are promising, but several central theoretical/interpretive claims are not currently supported with sufficiently clear or accurate evidence.

* **Estimand mismatch (ITE vs CATE):** The paper repeatedly refers to estimating "ITE," but the defined estimand and evaluation target is the conditional mean/dose–response $\mathbb{E}[Y(t)\mid X=x]$, i.e., a CATE. While earlier work (e.g., Shalit et al., 2017) used ITE terminology loosely, later work (e.g., Bellot et al., 2022) corrected this to CATE. This distinction matters because under ignorability only $\mathbb{E}[Y(t)\mid X=x]$ is identifiable, not unit-level ITE. The paper should correct the terminology or explicitly clarify its use of ITE.

* **Theory–implementation mismatch around Assumption 3 and compression/IB:** Assumption 3 requires the encoder to be a twice-differentiable bijection. Under a deterministic bijection, $I(X;Z)$ is constant and cannot be reduced, so IB/compression arguments cannot literally apply under that assumption. Yet Section 4.2/Theorem 3 motivate the method via controlling $I(X;Z)$ ("compression") and Section 5 implements compression heuristics (low-dimensional Z, capacity penalties). This creates a mismatch between the assumptions required for the theoretical bounds and the behavior of the implemented model.

    While similar bijectivity assumptions appear in prior work (e.g., Shalit et al., 2017; Bellot et al., 2022), there they are used as technical devices for reparameterization and distributional balancing, not to justify compression or information removal. Here, by contrast, bijectivity is invoked alongside IB-based compression arguments, creating a mismatch between the assumptions underlying the theory and the behavior of the implemented model.

* **Remark 1 ("tighter bound with CS") is not established as written:** This is the main theoretical differentiator emphasized in the abstract and in Remark 1: that replacing KL with CS yields a "tighter" counterfactual bound. As stated, the justification is not convincing for two reasons:
    * There is no general ordering between CS divergence and KL divergence. In general, for arbitrary distributions p and q, it is not true that $D_{CS}(p||q) \leq D_{KL}(p||q)$, nor the reverse. So the argument "the bound is tighter because CS is smaller than KL" cannot be made without additional assumptions.
    * The paper claims the $D_{CS} \leq D_{KL}$ inequality (Eq. (15)) holds approximately under "mild conditions" but those conditions are not clearly stated or easy to evaluate. The main paper points to Appendix B, but I could not find an explicit statement or sufficient conditions and proof steps that establish the claimed inequality. More broadly, for learned neural representations, $p(z,t)$ and $p(z)p(t)$ are implicit high-dimensional distributions, so any "mild conditions" should be spelled out in a way that makes them interpretable in this context. Otherwise the "tighter" claim reads more like a heuristic right now.

* **Theorem 3 statement form:** since it combines a high-probability generalization bound (Theorem 2) with the counterfactual bound, Theorem 3 should itself be high-probability (or expectation-over-samples); as written it reads deterministic and may be missing the probability qualifier/dependence on $\delta$.

* **Empirical reporting concerns that reduce confidence in Section 6’s conclusions**
	- TCGA improvements are unusually large (order-of-magnitude vs baselines), which could be real but deserves additional sanity checks/analysis.
	- The paired t-test procedure is underspecified (what is paired, how many runs, multiple comparisons).
	- Robustness plots appear to exclude TCGA; robustness would be stronger with broader baseline inclusion and an explicit overlap/support-mismatch stress test.

Overall, I believe the method and experiments are interesting, but key theoretical assumptions (e.g. bijectivity and differentiability)  and statements ("CS bounds tighter than KL" claims) and the connection between theory and the implemented algorithm require correction/clarification before the claims can be considered fully supported.

**Requested Changes:**

### Critical (required for acceptance recommendation)
1. **Resolve theory–implementation mismatch around Assumption 3 and IB:** clarify which theoretical results are intended to apply to bijective encoders vs the compressive/non-invertible encoders used in practice; revise claims accordingly.
2. **Correct/qualify the “CS tighter than KL” claim (Remark 1 / Eq. 15):** either provide explicit sufficient conditions and a clear derivation supporting the claimed inequality, or rephrase the claim to avoid implying a general dominance/ordering between CS and KL divergences.
3. **Fix Theorem 3 statement form:** add the appropriate probability qualifier (or state an expectation version), consistent with Theorem 2.
4. **TCGA sanity checks / analysis:** given the unusually large reported improvement, add additional diagnostics (more seeds/splits; error broken down by treatment region; or checks showing no artifact from synthetic construction).
5. **Clarify statistical testing protocol for paired t-test:** what is paired, how many runs, and whether any multiple-comparison correction is applied.
6. **Robustness experiments:** include TCGA (or explain why it’s excluded) and add at least one explicit overlap/support-mismatch stress test. Continuous treatment settings are particularly sensitive to weak overlap.

### Strengthening (not strictly required, but would improve the work)
7. **Fix estimand terminology:** replace "ITE" with "CATE / conditional dose–response" (or clearly define what is meant), and ensure metrics/definitions match throughout.
8. **Reconcile the MIMIC reporting inconsistency:** There is a mismatch between narrative text and Table 1 for MIMIC-IV (different reported $\sqrt{MISE}$), suggesting a metric/reporting mix-up. Ensure the metric definition (MISE vs $\sqrt{\text{MISE}}$) and the text and table values match.
9. **Minor fixes:** inconsistent notation (e.g., treatment vector notationin Eq. (9)), typo in Assumption 4 (should be F instead of G), and give more explanation/intuition around the eigenvalue condition in Theorem 1, Figure 1 left panel arrow between X and T is missing.

---

> ### Author Response · Authors · 2026-02-06
> **Rebuttal to Review by Reviewer sMnY**
>
> Many thanks for the extensive feedback. We appreciate that you find the work relevant and methodologically interesting.
>
> We will now continue with a clarification of the points of concern.
> ## Clarifications
>
> > Estimand mismatch
>
> We agree. Following Shalit et al.’s loose usage was imprecise. We have revised the manuscript to consistently use CATE instead of ITE throughout.
> * * *
> > Theory–implementation mismatch
>
> We thank the reviewer for this remark. We explicitly justified it by analogy to Kazemi and Ester (2024), who make a similar theoretical–empirical deviation. That said, we agree with the concern regarding the inconsistency. In the revision, we therefore propose the following:
>
> 1. We now clarify directly below Assumption 3 that it is a technical condition which is relaxed later.
>
> 2. We now state that while Assumption 3 implies a bijective encoder preserves mutual information, in practice the IB-inspired regularization functions as a capacity-control heuristic for a non-invertible encoder, explicitly avoiding phrasing such as “limiting $I(x; z)$ via the IB principle”.
> * * *
> > Remark 1 not established as written
>
> We thank you for the careful reading and for raising this important concern. We agree that there is no universal ordering between CS divergence and KL divergence for arbitrary distributions.
>
> We rephrased Remark 1 and clarified that the comparison holds only under explicitly stated bounded-domain and integrability assumptions. We now provide the precise conditions in Remark 1 and state the inequality with the associated constants $C_1$ and $C_2$, together with their definitions.
>
> * * *
> > Thm. 3 statement form
>
> We thank the reviewer for this important observation. We agree that, as originally written, Thm. 3 combines a high-probability generalization bound (Thm. 2) with a deterministic counterfactual bound, which makes the statement potentially ambiguous regarding probabilistic qualification.
>
> We removed the unified Thm. 3 (due to the mismatch assumption regarding encoders in Thm. 1 and Thm. 2) and instead present the counterfactual bound and the capacity-based generalization bound separately (Sections 4.1–4.2). The former is deterministic in nature, while the latter is stated as a high-probability bound with explicit dependence on $\delta$.
>
> We then (4.3) clarify how these independently established results jointly motivate the IBEX objective, without claiming a single unified probabilistic bound under non-invertible encoders.
>
> We believe this restructuring resolves the inconsistency noted by the reviewer and improves the clarity of the theoretical presentation.
> * * *
>
> > Empirical reporting concerns
>
> We reran ACFR and SCIGAN on TCGA using a small hyperparameter grid. SCIGAN improves to $\sqrt{\mathrm{MISE}} = 0.36$ (vs. $0.15$ for our method), while ACFR remains largely unchanged. This suggests that default hyperparameter choices contributed to SCIGAN’s weaker performance in this setting. We believe this reduces the performance gap and addresses your concern. We would be happy to increase the number of seeds or apply the same procedure to additional baselines if desired.
>
> > The paired t-test procedure underspecified
>
> We run 10 independent seeds and compute metrics on identical splits. We assess significance using paired t-tests comparing our method to each baseline across seeds, with Holm–Bonferroni correction. We clarify this in the caption of Table 1.
>
> > Robustness plots exclude TCGA
>
> We included TCGA in the robustness plot in the updated manuscript (see: Fig. 3).
> ## Requested Changes
> We will now address your requested changes.
>
> 1. > Resolve theory–implementation mismatch.
>
> See comment above.
> * * *
>
> 2. > Correct/qualify the “CS tighter than KL” claim
>
> We refer to our comment (regarding Remark 1) above.
> * * *
>
> 3. > Fix Thm. 3 statement form
>
> We refer to our comments on Thm. 3.
> * * *
>
> 4. > TCGA sanity checks/analysis
>
> We improved baseline hyperparameter search. See comment above.
> * * *
>
> 5. > Clarify statistical testing protocol
>
> See comment above.
> * * *
> 6. > include TCGA (or explain why it’s excluded) and add at least one explicit overlap/support-mismatch stress test
>
> We assess robustness to weak overlap using a support-mismatch stress test on TCGA covariates, training on treatments in $[0.2, 0.8]$ and evaluating over $[0, 1]$, requiring extrapolation beyond observed support. The results show a clear trade-off: IBEX performs best in-support, SCIGAN is most robust out-of-support, and ACFR degrades substantially under overlap violations. Under this stress test, SCIGAN achieves $\sqrt{\text{MISE}}$ of $0.36$ (In-Support), and $\mathbf{0.75}$ (Out-of-Support), while IBEX attains $\mathbf{0.20}$, and $1.06$, and ACFR records $0.84$, and $1.63$. Full details are in Appendix D.
> * * *
> 7. > Fix estimand terminology.
>
> See above.
>
> * * *
> 8. > MIMIC reporting inconsistency.
> 9. > Minor fixes.
>
> Typos and inconsistencies corrected.
>
> - Yu et al. 2024, ICLR.
> - Kazemi and Ester. 2024, AAAI.

---

> > ### Author Response · Authors · 2026-02-26
> > **Follow-up on Revised Manuscript and Rebuttal**
> >
> > Dear Reviewer sMnY,
> >
> > Thank you once again for your thorough and constructive review of our manuscript.
> >
> > We are writing to kindly follow up, as we have not yet received further comments. We hope that the revisions and improvements adequately addressed your initial concerns. However, if any issues remain or if additional clarification would be helpful, we would be more than happy to make further adjustments.
> >
> > Thank you again for your valuable input.
> >
> > Kind regards,
> >
> > The Authors

---

> > > ### Comment · Reviewer_sMnY · 2026-03-23
> > > **Thank you for the revisions**
> > >
> > > Thank you for the detailed response and for the substantial revisions. I appreciate you taking the time to address my concerns. In particular, I am glad to see:
> > >
> > > * the correction of the ITE/CATE terminology,
> > > * the clarification of Assumption 3 as a technical condition and the corresponding adjustment of the IB narrative,
> > > * the revision of Remark 1 to avoid implying a general ordering between CS and KL divergences,
> > > * the restructuring of the theoretical section (removal of Theorem 3 and separation of results),
> > > * the improvements to the empirical section, including clarified statistical testing, updated TCGA baselines, etc.
> > >
> > > These changes address my main concerns and significantly improve both the clarity and the agreement between the theoretical and empirical components of the paper. I have no additional comments at this point.

---

> > > > ### Author Response · Authors · 2026-03-23
> > > > **Thank you for your comment**
> > > >
> > > > Thank you very much for your reply. We are pleased to hear that our revisions and responses have addressed your main concerns.
> > > >
> > > > Should any additional questions or requests for clarification arise at a later stage, we would be happy to provide clarification.
> > > >
> > > > Kind regards,
> > > >
> > > > The authors

---

### Review · Reviewer_xjMt · 2026-01-08

**Summary Of Contributions:**

1.It derives a novel counterfactual generalization bound for continuous and multivariate treatment effect estimation based on Cauchy–Schwarz divergence, and shows that this bound is tighter than existing KL-based bounds under mild conditions.

2.It connects this bound with the Information Bottleneck principle, introducing a theoretically grounded representation-learning objective that jointly controls factual error and treatment–representation dependence without relying on variational or adversarial approximations.

3.It proposes IBEX, a practical neural architecture implementing these ideas, and demonstrates its effectiveness through extensive experiments on multiple benchmarks, including dose–response estimation, policy evaluation, robustness to treatment assignment bias, and ablation studies.

**Audience:**

Yes

**Audience Explanation:**

The paper should interest parts of the TMLR audience working on causal inference and representation learning, as it addresses continuous and multivariate treatment effect estimation using a simple, non-adversarial, information-theoretic approach that performs well in this more general setting.

**Claims And Evidence:**

Yes

**Claims Explanation:**

The empirical claims are generally well supported. The paper evaluates counterfactual performance using standard metrics (MISE and PE) on multiple benchmarks with continuous treatments, following established practice by relying on synthetically generated outcomes to obtain ground-truth counterfactuals. The inclusion of strong baselines, robustness analyses, and ablations provides convincing empirical evidence, though the counterfactual evaluation protocol could be stated more explicitly.

**Requested Changes:**

1.Overlap assumption: For continuous treatments, the overlap (positivity) assumption should be stated in terms of the conditional treatment density(PDF) being strictly positive (almost everywhere), rather than probabilities or CDF-based statements, which are not well-defined in the continuous case.

2.Assumption 3 requires the encoder to be bijective, which is strong and at odds with the compressive representation-learning objective; it would be helpful for the authors to clarify how the counterfactual generalization bound would change if this assumption were relaxed, and to discuss how the theory aligns with the existence or learnability of treatment-invariant representations in more realistic, non-invertible settings.

3.While the tighter CS-based bound provides useful theoretical motivation, it would be helpful to clarify whether any empirical evidence can be provided in a simple synthetic setting to illustrate the relationship between the bound and counterfactual error, and to explicitly note that a tighter theoretical bound does not necessarily imply improved empirical performance.

---

> ### Author Response · Authors · 2026-02-06
> **Rebuttal to Review by Reviewer xjMt**
>
> First, we sincerely thank the reviewer for the thoughtful feedback. We are glad that you find the work interesting and the claims well supported.
>
> ## Requested Changes
>
> We will now address your concerns/requested changes:
>
> 1. > Overlap assumption.
>
> This is a fair point. In the revised manuscript (revision can be found above), we restated the assumption as follows:
>
> **Assumption 2.** We assume that the conditional treatment distribution admits a density $p_{\mathbf{T}_{\mathrm{f}} \mid \mathbf{X}}(\mathbf{t} \mid \mathbf{x})$ with respect to the Lebesgue measure,
>
> and that this density is strictly positive almost everywhere on $\mathcal{X} \times \mathcal{T}$,
> i.e., $p_{\mathbf{T}_{\mathrm{f}} \mid \mathbf{X}}(\mathbf{t} \mid \mathbf{x}) > 0
> \quad \text{for almost every } (\mathbf{x}, \mathbf{t}) \in \mathcal{X} \times \mathcal{T}.$
>
>
> * * *
>
> 2. > Assumption 3 requires the encoder to be bijective
>
> Thank you for pointing this out. We appreciate this suggestion and have clarified the point. This issue was also raised by Reviewer sMnY, and we refer to our detailed response there. In the revised manuscript, we now discuss in more detail the complementary roles of dependence and capacity in an additional subsection (Subsection 4.3), and explain how these independently established results jointly motivate the IBEX objective. In addition, we present the counterfactual bound and the capacity-based generalization bound separately (Sections 4.1–4.2), rather than as a unified theorem.
>
> * * *
>
> 3a. > clarify whether any empirical evidence can be provided in a simple synthetic setting to illustrate the relationship between the bound and counterfactual error
>
> We thank the reviewer for the useful suggestion. In Appendix D, we included an additional synthetic experiment that further illustrates this relationship by varying the strength of confounding while holding the learning procedure fixed. The key takeaway is that the proposed bound behaves as a meaningful upper-control quantity: as counterfactual error increases with confounding, the bound increases monotonically and remains tighter under the CS divergence than under KL, without implying any change in empirical performance.
>
> * * *
>
> 3b. > explicitly note that a tighter theoretical bound does not necessarily imply improved empirical performance.
>
> We acknowledge this explicitly now, stating (directly after Remark 1 in the revised manuscript) that:
>
> *It is important to note that a tighter bound does not necessarily imply improved empirical performance; rather, it guarantees only that, under the assumptions of the theory, the Cauchy-Schwarz divergence yields a tighter theoretical upper bound on counterfactual error than the KL divergence.*

---

### Review · Reviewer_oYiS · 2026-01-24

**Summary Of Contributions:**

This work aimed to improve the algorithm for continuous and multivariate treatments. The contribution includes the error boound based on the Cauchy-Schwarz (CS) divergence and the information bottleneck principle.

**Audience:**

Yes

**Audience Explanation:**

While the Kullback–Leibler (KL) divergence is popularly applied, some researchers should be interested at the study of Cauchy-Schwarz (CS) divergence.

**Claims And Evidence:**

Yes

**Claims Explanation:**

The claims look fine.

**Requested Changes:**

1. Why are Assumptions 1 and 2 reasonable and necessary?
2. Is the bound depending on the CS divergence better than the one based on the KL divergence? Some comparison on the theoretical results is appreciated.

---

> ### Author Response · Authors · 2026-02-06
> **Rebuttal to Review by Reviewer oYiS**
>
> We thank the reviewer for taking the time to review our work and for the insightful questions.
>
> ## Requested Changes
>
>
> We will now address your concerns/requested changes.
>
> 1. > Why are Assumptions 1 and 2 reasonable and necessary?
>
> Assumptions 1 and 2 are standard and minimal requirements for causal identification with observational data (e.g., Shalit et al., 2017; Pearl, 2000). They are theoretically necessary for our counterfactual bounds to hold and practically reasonable in the rich-data regimes and continuous-treatment applications we target. Importantly, our method does not strengthen these assumptions beyond what is already required by existing approaches, but instead focuses on improving robustness and generalization under the same causal conditions as existing methodologies.
>
> * * *
>
> 2. > Is the bound depending on the CS divergence better than the one based on the KL divergence? Some comparison on the theoretical results is appreciated.
>
> Assuming that “better” refers to a tighter bound, then under specific conditions (which we now clarify explicitly in Remark 1 in the revised manuscript), the answer is yes. A similar concern was raised by Reviewer sMnY. Specifically, in Remark 1 we now dedicate an additional page to formally outlining the conditions under which the CS-divergence–based bound is tighter than the KL-based bound, and we provide an explicit proof based on Yu et al. (2024) and Yin et al. (2024).
>
> At the request of Reviewer xjMt, we also include an additional experiment in Appendix D that directly compares the two bounds in a simplified setting where the no-hidden-confounding assumption (Assumption 1 in the manuscript) is violated.
> Intuitively, the CS-based bound is preferable because it measures dependence between the representation and treatment more smoothly than KL. Since KL can become arbitrarily large due to small mismatches in low-density regions, CS typically yields a smaller penalty and thus a tighter bound.
>
> Sources:
> - Yu et al. 2024, ICLR.
> - Shalit et al. 2017, ICML.
> - Pearl, *Causality*

---

### Author Response · Authors · 2026-02-06
**Global Comment to All Reviewers**

# Global Comment

We sincerely thank all reviewers for their valuable feedback, thoughtful questions, and the time they dedicated to reviewing our work.

We were encouraged by the following positive feedback:

1. The relevance of our work to (continuous) causal inference (all reviewers).
2. The suitability of using the Cauchy–Schwarz divergence (all reviewers).
3. Strong support for our claims through empirical results and theory (Reviewers xjMt and oYiS).

We also identified several valuable points for improvement:

1. Considerations regarding encoder bijectivity versus non-bijectivity in practice (Reviewers xjMt and sMnY).
2. A more thorough comparison between the KL bound and the CS bound (all reviewers).

We address these points in detail in the individual responses below. In addition to our rebuttal comments, we have uploaded a revised manuscript incorporating these changes. We are happy to make further improvements if needed.

---

### Author Response · Authors · 2026-04-11
**Thanks to the reviewers**

Dear Reviewers,

We were pleased to receive the decision and would like to thank you for the time and effort you invested in evaluating our work. We greatly appreciate your interest in our research, and your insightful comments and questions have helped us improve the manuscript.

Kind regards,

The authors

---

### Decision · Action_Editor_L5gz · 2026-04-02

**Recommendation:** Accept as is

**Audience:**

Yes

**Audience Explanation:**

The paper addresses a common machine learning problem and is well connected to previous literature. It has a clear technical contribution, a new generalization bound, which is likely to be interesting at least for people working specifically on continuous treatment effect estimation. The paper is fairly theoretical in nature and the audience is somewhat limited, but not extremely small. Finally, the empirical performance of the method is good and this may be of interest for practitioners, with the method possibly becoming a strong baseline at least.

**Claims And Evidence:**

Yes

**Claims Explanation:**

The paper provides a new theoretical generalization bound for continuous treatment effect estimation and presents a computational method that is demonstrated to work well on benchmark data sets. The accuracy is claimed to be state-of-the-art, which is a strong claim, and robust under treatment-assignment bias.

The approach is well justified from a theoretical perspective, building directly on a new generalization bound that is proven formally. The empirical validation is sufficient and the claim of state-of-the-art performance is justified, with clear improved performance over previous methods on commonly used benchmarks. The robustness is also directly verified. Overall, the claims are hence justified both theoretically and empirically, and the reviewers did not point out issues with the evidence.

---

> ### Author Response · Authors · 2026-04-11
> **Camera Ready Version Submitted**
>
> Dear Arto Klami,
>
> Many thanks for the notification and for taking the time to assess and edit our work. We are pleased to hear the manuscript has been accepted.
>
> We have now uploaded the camera-ready version. In addition to de-anonymizing the author names and references to the online code repository, we have also added a brief acknowledgment section at the end of the main text. The remainder of the manuscript has not been changed.
>
> Please let us know if any further modifications are required.
>
> Kind regards,
>
> The authors